# FrameBridge: Improving Image-to-Video Generation with Bridge Models

Yuji Wang [* 1]  Zehua Chen [* 1 2]  Xiaoyu Chen [1]  Yixiang Wei [1]  Jun Zhu [1 2]  Jianfei Chen [1]

## Abstract

Diffusion models have achieved remarkable progress on image-to-video (I2V) generation, while their noise-to-data generation process is inherently mismatched with this task, which may lead to suboptimal synthesis quality. In this work, we present FrameBridge. By modeling the *frame-to-frames* generation process with a bridge model based *data-to-data* generative process, we are able to fully exploit the information contained in the given image and improve the consistency between the generation process and I2V task. Moreover, we propose two novel techniques toward the two popular settings of training I2V models, respectively. Firstly, we propose *SNR-Aligned Fine-tuning (SAF)*, making the first attempt to fine-tune a diffusion model to a bridge model and, therefore, allowing us to utilize the pre-trained diffusion-based text-to-video (T2V) models. Secondly, we propose *neural prior*, further improving the synthesis quality of FrameBridge when training from scratch. Experiments conducted on WebVid-2M and UCF-101 demonstrate the superior quality of FrameBridge in comparison with the diffusion counterpart (zero-shot FVD 95 vs. 192 on MSR-VTT and non-zero-shot FVD 122 vs. 171 on UCF-101), and the advantages of our proposed SAF and neural prior for bridge-based I2V models. The project page: https://framebridge-icml.github.io/.

## 1 Introduction

Image-to-video (I2V) generation, commonly referred as image animation, aims at generating consecutive video frames from a static image (Xing et al., 2024; Ni et al., 2023; Zhang et al., 2024a; Guo et al., 2024; Hu et al., 2022), *i.e.*, a *frame-to-frames* generation task where maintaining appearance consistency and ensuring temporal coherence of generated video frames are the key evaluation criteria (Xing et al., 2024; Zhang et al., 2024a). With the recent progress in video synthesis (Brooks et al., 2024; Yang et al., 2024b; Blattmann et al., 2023; Bao et al., 2024), several diffusion-based I2V frameworks have been proposed, with novel designs on network architecture (Xing et al., 2024; Zhang et al., 2024a; Chen et al., 2023b; Ren et al.; Lu et al.), cascaded framework (Jain et al., 2024; Zhang et al., 2023), and motion representation (Zhang et al., 2024b; Ni et al., 2023). However, although these methods have demonstrated the strong capability of diffusion models (Ho et al., 2020; Song et al.) for I2V synthesis, their *noise-to-data* generation process is inherently mismatched with the *frame-to-frames* synthesis of I2V task, making them suffer from the difficulty of generating high-quality video samples from uninformative Gaussian noise rather than the given image.

In this work, inspired by recently proposed bridge models (Chen et al.; Liu et al., 2023; Chen et al., 2023c), we present FrameBridge, a novel I2V framework to model the *frame-to-frames* synthesis process with a *data-to-data* generative framework instead of the *noise-to-data* one in diffusion models. Specifically, given the input image and video target, we first leverage variational auto-encoder (VAE) based compression network to transform them into continuous latent representations, and then take their latent representations as boundary distributions, *i.e.*, prior and target, to establish our *data-to-data* generative framework. Considering the static image has already been an informative prior for each of the consecutive frames in video target, we naturally replicate it to obtain the prior of the whole video clip, constructing the *frames-to-frames* training data pairs for the prior-to-target generative framework in FrameBridge. Standing on constructed pairs, we establish bridge models (Tong et al., 2024; Zhou et al.; Chen et al., 2023c) between them to learn the I2V synthesis with Stochastic Differential Equation (SDE) based generation process. In comparison with previous diffusion-based I2V methods, our FrameBridge utilizes given static image as the prior of video target, which is advantageous on preserving the appearance details of input image than conditionally generating video samples from random noise. Moreover, our frames-to-frames bridge model learns image animation in model training rather than learn-

---

*Equal contribution [1]Dept. of Comp. Sci. and Tech., Institute for AI, BNRist Center, THBI Lab, Tsinghua-Bosch Joint ML Center, Tsinghua University [2]ShengShu, Beijing, China. Correspondence to: Jianfei Chen <jianfeic@tsinghua.edu.cn>.

*Proceedings of the 42nd International Conference on Machine Learning*, Vancouver, Canada. PMLR 267, 2025. Copyright 2025 by the author(s).

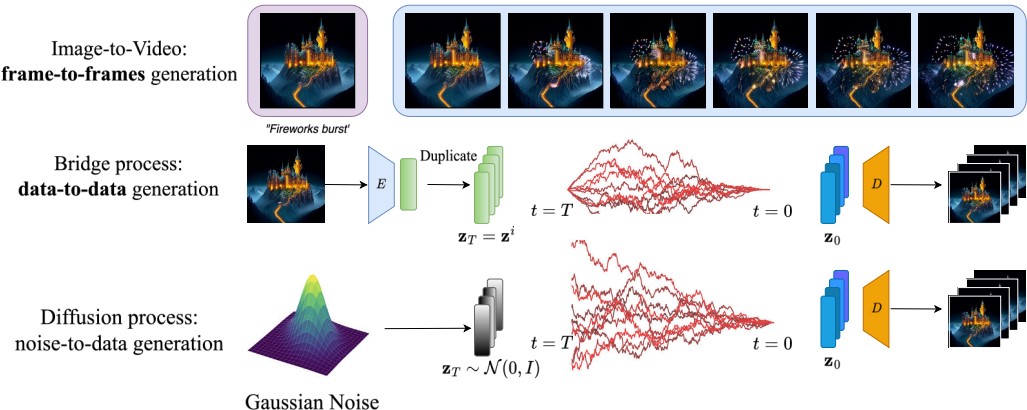

*Figure 1.* **Overview of FrameBridge and diffusion-based I2V models.** The sampling process of FrameBridge (upper) starts from given static image, while diffusion models (lower) synthesize videos from uninformative Gaussian noise.

ing image-conditioned noise-to-video generation, which enhances the consistency between generative framework and I2V task, *i.e.*, *data-to-data* for *frame-to-frames* and tends to benefit temporal coherence for I2V synthesis.

In practice, I2V systems usually take advantage of a pre-trained diffusion-based text-to-video (T2V) model (Xing et al., 2024; Chen et al., 2023b; Ma et al., 2024a) with a fine-tuning process, to reduce the requirements of image-video data pairs and the computational resources at the training stage of I2V generation. Toward efficiently utilizing previously pre-trained *diffusion-based T2V models*, we propose SNR-Aligned Fine-tuning (SAF), a novel technique for fine-tuning them to *bridge-based I2V models*. Specifically, we first reparameterize the bridge process in Frame-Bridge, enabling the noisy intermediate representations of our frames-to-frames process to be aligned with the ones in the noise-to-frames process of pre-trained diffusion models, improving fine-tuning efficiency. Then, we change the timestep to match the signal-to-noise (SNR) ratio between the input of the bridge model and the pre-trained diffusion model, remaining the differences between the diffusion and bridge process. Our SAF aligns the noisy intermediate representations of two generative frameworks while preserving the difference between them (*i.e.*, diffusion and bridge process), and therefore improves the final synthesis quality of FrameBridge when adapting pre-trained T2V diffusion models.

Compared to diffusion models using Gaussian prior, Frame-Bridge takes the given static image as the prior of video target to improve I2V performance. Toward further improving bridge-based I2V synthesis quality, we present a stronger prior for FrameBridge. Given a static image, we design a one-step mapping-based network and optimize it with the video target, extracting a *neural prior* from the image for the video target. Compared to input image, this

neural prior reduces the distance between prior and video target to a greater extent, and alleviates the burden of generation process further. Although more advanced methods can be leveraged to extract more informative neural prior, we empirically find that a coarse estimation for video target at the cost of a single sampling step has already been beneficial to FrameBridge. This further verifies our motivation to present FrameBridge and shows a novel method to enhance bridge-based I2V models. In this work, our contributions can be summarized as follows:

- We propose FrameBridge, making the first attempt to model the frame-to-frames generation task of I2V with a data-to-data generative framework.

- We present two novel techniques, SAF and neural prior, further improving the performance of FrameBridge when fine-tuning from pre-trained T2V diffusion models and training from scratch respectively.

- We conduct experiments on two I2V benchmarks by training FrameBridge on WebVid-2M (Bain et al., 2021) and UCF-101 (Soomro, 2012). Compared with its diffusion counterpart, FrameBridge fine-tuned with SAF reduces the zero-shot FVD (Unterthiner et al. (2018); lower is better) from 192 to 95 on MSR-VTT (Xu et al., 2016), and FrameBridge with neural prior trained from scratch reduces the non-zero-shot FVD from 171 to 122 on UCF-101, highlighting the superiority of FrameBridge to their diffusion counterparts and the effectiveness of SAF and neural prior.

## 2  Related Works

**Diffusion-based I2V Generation**    Diffusion models have recently achieved remarkable progress in I2V synthesis (Blattmann et al., 2023; Chen et al., 2023a; Li et al.,

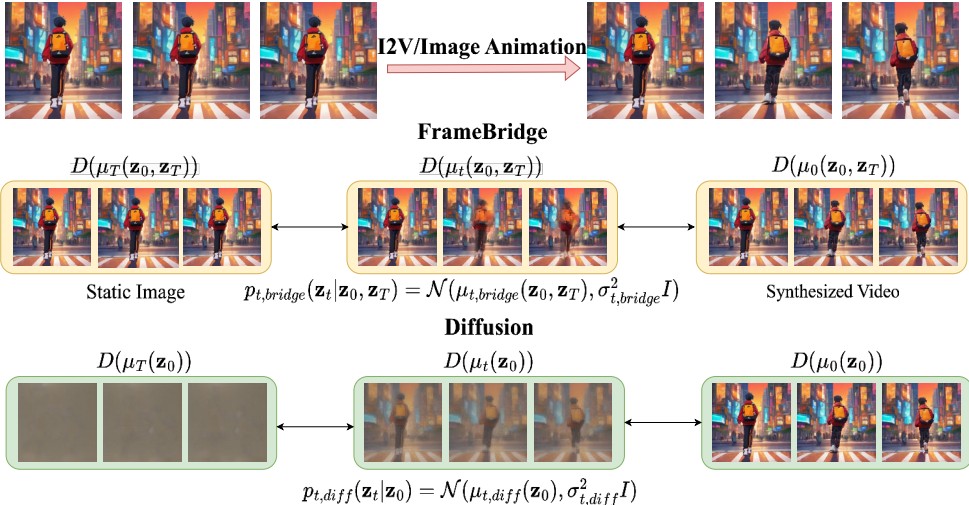

**FrameBridge**

$D(\mu_T(\mathbf{z}_0, \mathbf{z}_T))$          $D(\mu_t(\mathbf{z}_0, \mathbf{z}_T))$          $D(\mu_0(\mathbf{z}_0, \mathbf{z}_T))$

Static Image    $p_{t,bridge}(\mathbf{z}_t|\mathbf{z}_0, \mathbf{z}_T) = \mathcal{N}(\mu_{t,bridge}(\mathbf{z}_0, \mathbf{z}_T), \sigma^2_{t,bridge}I)$    Synthesized Video

**Diffusion**

$D(\mu_T(\mathbf{z}_0))$          $D(\mu_t(\mathbf{z}_0))$          $D(\mu_0(\mathbf{z}_0))$

$p_{t,diff}(\mathbf{z}_t|\mathbf{z}_0) = \mathcal{N}(\mu_{t,diff}(\mathbf{z}_0), \sigma^2_{t,diff}I)$

*Figure 2.* **Visualization for the mean value of marginal distributions**. We visualize the decoded mean value of bridge process and diffusion process. The prior and target of FrameBridge are naturally suitable for I2V synthesis.

2024) and proposed multi-stage generation system (Jain et al., 2024; Zhang et al., 2023; Shi et al., 2024; Zhang et al., 2025f), fusion module (Wang et al., 2024; Ren et al.) and improved network architectures (Wang et al., 2024; Xing et al., 2024; Ma et al., 2024a; Chen et al., 2023b; Ren et al.; Zhang et al., 2025a;b;c;d). However, their *noise-to-data* generation process may be inefficient for I2V synthesis. To improve the uninformative prior of diffusion models (Fischer et al., 2023; Albergo et al., 2024; Yang et al., 2024a), PYoCo (Ge et al., 2023) proposes to use correlated noise for each frame in both training and inference. ConsistI2V (Ren et al.), FreeInit (Wu et al., 2024), and CIL (Zhao et al.) present training-free strategies to better align the training and sampling distribution of diffusion prior. These strategies improve the noise distribution to enhance the quality of synthesized videos, while they still suffer the restriction of noise-to-data diffusion framework, which may limit their endeavor to utilize the entire information (*e.g.*, both large-scale features and fine-grained details) contained in the given image. In this work, we propose a *data-to-data* framework and utilize clean and deterministic prior rather than Gaussian noise, allowing us to leverage the given image as prior information.

**Bridge Models** Recently, bridge models (Chen et al.; Tong et al., 2024; Liu et al., 2023; Zhou et al.; Chen et al., 2023c; Zheng et al., 2024; He et al.; De Bortoli et al., 2021; Peluchetti, 2023), which overcome the restriction of Gaussian prior in diffusion models, have gained increasing attention. They have demonstrated the advantages of *data-to-data* generation process over the *noise-to-data* one on image-to-image translation (Liu et al., 2023; Zhou et al.) and speech synthesis (Chen et al., 2023c; Li et al., 2025) tasks. In this work, we make the first attempt to extend

bridge models to I2V synthesis and further propose two improving techniques for bridge models, enabling efficient fine-tuning from diffusion models and stronger prior for video target.

## 3 Motivation

**Diffusion-based I2V Synthesis** I2V synthesis aims at generating a video clip $\mathbf{v} \in \mathbb{R}^{L \times H \times W \times 3}$ with $L$ frames conditioning on a static image, *e.g.*, the initial frame $v^i \in \mathbb{R}^{H \times W \times 3}$ of video clip $\mathbf{v}$. In diffusion-based I2V systems (Xing et al., 2024; Blattmann et al., 2023), an VAE-based compression network is usually leveraged to first transform the video $\mathbf{v}$ into a latent $\mathbf{z} \in \mathbb{R}^{L \times h \times w \times d}$ in a per-frame manner with a pre-trained image encoder $\mathcal{E}(\mathbf{v})$, where $h = \frac{H}{p}$, $w = \frac{W}{p}$, $p > 1$ and $d$ are the spatial compression ratio and the number of output channels. A forward diffusion process gradually converts the video latent $p_0(\mathbf{z}_0|z^i, c) \triangleq p_{data}(\mathbf{z}_0|z^i, c)$ to a known prior distribution $p_{T,diff}(\mathbf{z}_T) \triangleq p_{prior,diff}(\mathbf{z}_T)$ with a forward SDE (Song et al.):

$$d\mathbf{z}_t = \boldsymbol{f}(t)\mathbf{z}_t dt + g(t)d\mathbf{w}, \quad \mathbf{z}_0 \sim p_{data}(\mathbf{z}_0|z^i, c), \quad (1)$$

where $\mathbf{w}$ is a Wiener process, $\boldsymbol{f}$ and $g$ are known coefficients, $z^i \in \mathbb{R}^{h \times w \times d}$ is the compressed latent of the initial frame $v^i$, and $c$ denotes other guidance such as the text prompt (Ma et al., 2024a; Chen et al., 2023b) or the class condition (Ni et al., 2023; Zhang et al., 2024b). In sampling, we first synthesize the latent $\mathbf{z} \sim p_0(\mathbf{z}_0|z^i, c)$ with the backward SDE which shares the same marginal distribution $p_{t,diff}(\mathbf{z}_t|z^i, c)$ (Song et al.):

$$\mathrm{d}\mathbf{z}_t = \left[ \boldsymbol{f}(t)\mathbf{z}_t - g(t)^2 \nabla_{\mathbf{z}_t} \log p_{t,diff}(\mathbf{z}_t | z^i, c) \right] \mathrm{d}t \\ + g(t)\mathrm{d}\bar{\mathbf{w}}, \quad \mathbf{z}_T \sim p_{prior,diff}(\mathbf{z}_T), \quad (2)$$

from a Gaussian prior $p_{prior,diff}(\mathbf{z}_T) \sim \mathcal{N}(0, I)$, and then decode the video clip with pre-trained VAE decoder $\mathcal{D}(\mathbf{z})$. To estimate the score function $\nabla_{\mathbf{z}_t} \log p_{t,diff}(\mathbf{z}_t | z^i, c)$, a U-Net (Ronneberger et al., 2015; Ho et al., 2020) or DiT (Peebles & Xie, 2023; Bao et al., 2023) based neural network is optimized with a denoising objective:

$$\mathcal{L}(\theta) = \mathbb{E}_{(\mathbf{z}_0, z^i, c), t, \mathbf{z}_t} \left[ \lambda(t) \left\| \boldsymbol{\epsilon}_\theta(\mathbf{z}_t, t, z^i, c) - \frac{\mathbf{z}_t - \alpha_t \mathbf{z}_0}{\sigma_t} \right\|^2 \right], \quad (3)$$

Here $\lambda(t)$ is a time-dependent weight function, and $\boldsymbol{\epsilon}_\theta(\mathbf{z}_t, t)$ is an alternative parameterization method of the score function (Ho et al., 2020).

**Limitations** As shown, the forward process of diffusion models gradually injects noise into data samples, which results in a boundary distribution at $t = T$ sharing the same distribution with the injected noise, *e.g.*, the standard Gaussian noise $\boldsymbol{\epsilon} \sim \mathcal{N}(\mathbf{0}, \boldsymbol{I})$. Therefore, in generation, their sampling process has to start from the uninformative prior distribution $p_{prior,diff}(\mathbf{z}_T) \sim \mathcal{N}(\mathbf{0}, \boldsymbol{I})$ and then iteratively synthesize the video latent $\mathbf{z}_0$ with learned conditional score function $\nabla_{\mathbf{z}_t} \log p_t(\mathbf{z}_t | z^i, c)$.

However, for I2V generation, the two key requirements are preserving the appearance details of the given static image (Ren et al.; Ma et al., 2024a) and ensuring temporal coherence between generated video frames (Guo et al., 2024; Zhang et al., 2024c). The noise prior of diffusion models and the mismatch between *noise-to-data* generation and *frame-to-frames* synthesis inevitably increase the burden of the generation process when meeting these two requirements. In this work, we propose FrameBridge. By modeling I2V with a *data-to-data* process, we simultaneously improve the prior of generation process for preserving the appearance details and enhance the consistency between the generative framework and I2V task for ensuring temporal coherence, leading to improved I2V performance.

## 4 FrameBridge

### 4.1 Bridge-based I2V Synthesis

Considering the given image, *i.e.*, initial frame $z^i$, has provided the appearance details and the starting point of animation for video target, we take it as the prior of following frames. To construct the boundary distributions for bridge models, we replicate the image latent $z^i$ for $L$ times along

temporal axis to obtain $\mathbf{z}^i \in \mathbb{R}^{L \times h \times w \times d}$ as the prior of video latent $\mathbf{z} \in \mathbb{R}^{L \times h \times w \times d}$, and establish the bridge process as follows.

**Bridge Process** In Figure 1, we present the overview of FrameBridge and compare it with diffusion-based I2V generation. Different from diffusion-based I2V models using uninformative Gaussian prior, our FrameBridge replaces the Gaussian prior with a Dirac prior $\delta_{\mathbf{z}^i}$, building a bridge process (Zhou et al.) to connect the video target and the replicated image prior $p_{prior,bridge}(\mathbf{z}_T | z^i, c) \triangleq \delta_{\mathbf{z}^i}(\mathbf{z}_T)$. Specifically, the forward process is changed from Equation (1) in diffusion models to:

$$\mathrm{d}\mathbf{z}_t = \left[ \boldsymbol{f}(t)\mathbf{z}_t + g(t)^2 \boldsymbol{h}(\mathbf{z}_t, t, \mathbf{z}_T, z^i, c) \right] \mathrm{d}t \\ + g(t)\mathrm{d}\boldsymbol{w}, \quad \mathbf{z}_0 \sim p_{data}(\mathbf{z}_0 | z^i, c), \quad \mathbf{z}_T = \mathbf{z}^i, \quad (4)$$

where $\boldsymbol{h}(\mathbf{z}_t, t, \mathbf{z}_T, z^i, c) \triangleq \nabla_{\mathbf{z}_t} \log p_{T,diff}(\mathbf{z}_T | \mathbf{z}_t)$ and $p_{T,diff}(\mathbf{z}_T | \mathbf{z}_t)$ is the marginal distribution of diffusion process shown in Equation (1). For bridge process, we denote the marginal distribution of Equation (4) as $p_{t,bridge}(\mathbf{z}_t | z^i, c)$. Similar to the forward SDE Equation (1) in diffusion process, the forward process of bridge models Equation (4) also has a reverse process, which shares the same marginal distribution $p_{t,bridge}(\mathbf{z}_t | z^i, c)$ and can be represented by the backward SDE:

$$\mathrm{d}\mathbf{z}_t = [\boldsymbol{f}(t)\mathbf{z}_t - g(t)^2 (\boldsymbol{s}(\mathbf{z}_t, t, \mathbf{z}_T, z^i, c) \\ - \boldsymbol{h}(\mathbf{z}_t, t, \mathbf{z}_T, z^i, c))]\mathrm{d}t + g(t)\mathrm{d}\bar{\boldsymbol{w}}, \quad \mathbf{z}_T = \mathbf{z}^i, \quad (5)$$

where $\boldsymbol{s}(\mathbf{z}_t, t, \mathbf{z}_T, z^i, c) \triangleq \nabla_{\mathbf{z}_t} \log p_{t,bridge}(\mathbf{z}_t | \mathbf{z}_T, z^i, c)$.

The change from the diffusion to the bridge process removes the restriction of noisy prior, allowing the generation process to start from a static image rather than previous Gaussian noise. Moreover, as the perturbation kernel $p_{t,bridge}(\mathbf{z}_t | \mathbf{z}_0, \mathbf{z}_T, z^i, c)$ in bridge process remains Gaussian (Appendix A), it facilitates us to find connections between the marginal distribution, *i.e.*, the intermediate representations of diffusion and bridge process, and then leverage the power of pre-trained diffusion models for bridge models.

**Training Objective** Analogous to diffusion models, we use a SDE solver to solve Equation (5) when sampling videos. Since $\boldsymbol{h}(\mathbf{z}_t, t, \mathbf{z}_T, z^i, c)$ can be calculated analytically (see Appendix A), we only need to estimate the unknown term $\boldsymbol{s}(\mathbf{z}_t, t, \mathbf{z}_T, z^i, c)$ with neural networks (Kingma et al., 2021). After parameterization as shown in Appendix A, we train our models $\boldsymbol{\epsilon}_\theta^{\check{\Psi}}(\mathbf{z}_t, t, \mathbf{z}_T, z^i, c)$ with the denoising objective (Chen et al., 2023c):

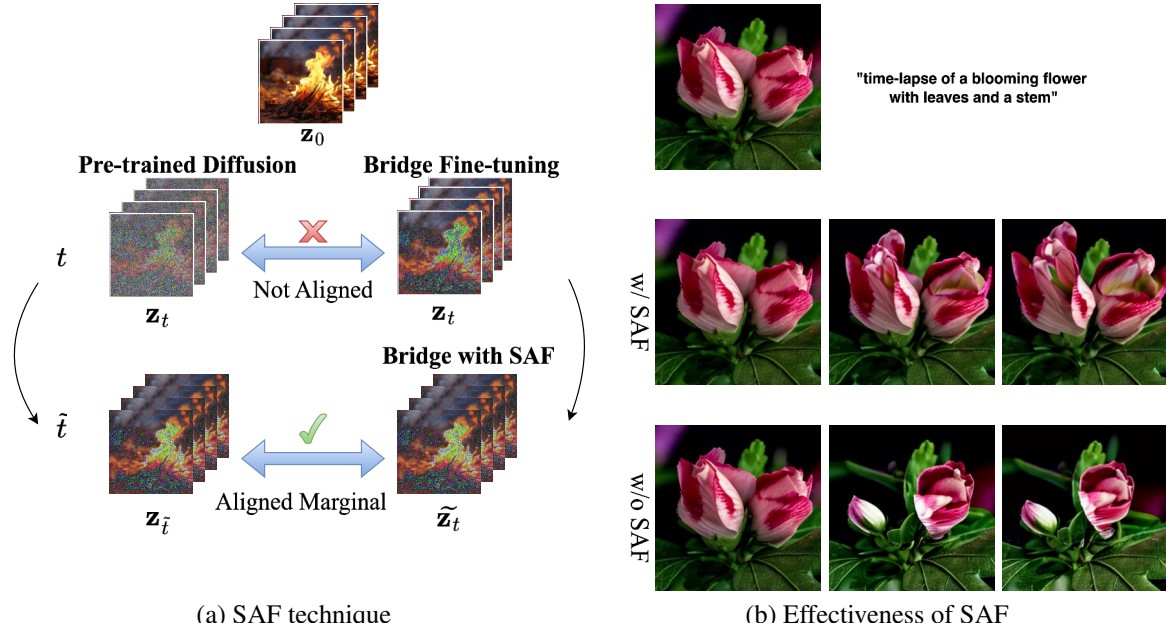

(a) SAF technique        (b) Effectiveness of SAF

*Figure 3.* **SNR-Aligned Fine-tuning for FrameBridge.** (a) SAF technique aligns the noisy latents of bridge process and diffusion process with respective timesteps, enabling efficient fine-tuning from diffusion-based T2V model to bridge-based I2V models. (b) FrameBridge with SAF can better leverage the capability of pre-trained models.

$$\mathcal{L}_{bridge}(\theta) = \mathbb{E}_{\substack{(\mathbf{z}_0, z^i, c) \sim p_{data}(\mathbf{z}_0, z^i, c), \mathbf{z}_T = \mathbf{z}^i, \\ t, \mathbf{z}_t \sim p_{t,bridge}(\mathbf{z}_t | \mathbf{z}_0, \mathbf{z}_T, z^i, c)}}$$
$$\left[ \left[ \left\| \boldsymbol{\epsilon}_\theta^{\hat{\Psi}}(\mathbf{z}_t, t, \mathbf{z}_T, z^i, c) - \frac{\mathbf{z}_t - \alpha_t \mathbf{z}_0}{\sigma_t} \right\|^2 \right] \right]. \tag{6}$$

The training of FrameBridge resembles that of Gaussian diffusion-based I2V models: We first sample a video latent $\mathbf{z}_0$ and the condition $c$ from training set, extracting the first frame of $\mathbf{z}_0$ to construct $\mathbf{z}^i$. The primary difference lies in the Gaussian perturbation kernel $p_{t,bridge}(\mathbf{z}_t | \mathbf{z}_0, \mathbf{z}_T, z^i, c)$ of Equation (6). As we replace the Gaussian prior with a deterministic representation $\mathbf{z}_T$, the mean value is an interpolation between data and $\mathbf{z}_T$ instead of the decaying data in diffusion models, naturally preserving more data information and facilitating generative models to learn image animation rather than regenerating the information provided in static image.

**Bridge Process vs Diffusion Process** To demonstrate the advantages of bridge process in I2V synthesis, we visualize the data part, *i.e.*, the mean function of bridge and diffusion process, in Figure 2. As shown, when replicating the initial frame, I2V synthesis can be formulated as a *frames-to-frames* generation task. With the *data-to-data* bridge process, the boundary distributions of our FrameBridge have been an ideal fit for the I2V task, which is helpful for generative models to focus on modeling the image animation

process.

In the meanwhile, as seen from our intermediate representations, the data information, *e.g.*, appearance details, is well preserved during the bridge process. In comparison, the prior and intermediate representations of diffusion process contain rare or coarse information of the target, which is uninformative and requires diffusion models to generate entire video information from scratch.

### 4.2 Efficient Fine-tuning

A common practice of training I2V models is to fine-tune from pre-trained T2V diffusion models (Chen et al., 2023b;a; Xing et al., 2024; Blattmann et al., 2023; Ma et al., 2024a). The essential difference between the diffusion and bridge process lies in the distribution of noisy latents $\mathbf{z}_{t,diff} \sim p_{t,diff}(\mathbf{z}_t)$ and $\mathbf{z}_{t,bridge} \sim p_{t,bridge}(\mathbf{z}_t)$. For certain $t \in [0, T]$, the pre-trained diffusion models only have the capability to denoise $\mathbf{z}_{t,diff}$ while our fine-tuning target is to denoise $\mathbf{z}_{t,bridge}$, and the substantial discrepancy between noisy latents makes it difficult to utilizing knowledge of pre-trained models. To address this issue, we believe that aligning the latents will allow us to fully leverage the denoising capability and learned representations of pre-trained models, which is critical to a more efficient and effective fine-tuning process (Yu et al., 2024a). Thus, we propose the innovative SNR-Aligned Fine-tuning (SAF) technique to align the latent $\mathbf{z}_{t,bridge}$ with a diffusion noisy latent $\mathbf{z}_{\tilde{t},diff}$. Note that we use a different timestep $\tilde{t} \neq t$, and we

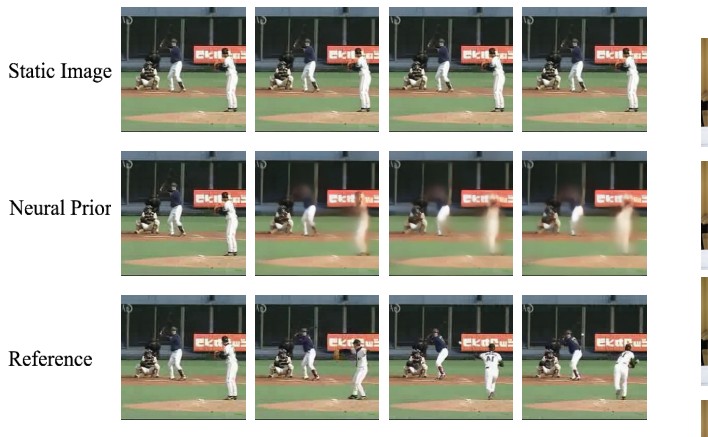

*Figure 4.* **Case of neural prior.** Our neural prior provides more motion information than the given static image, and is intuitively closer to the video target, further improving the prior of generation process.

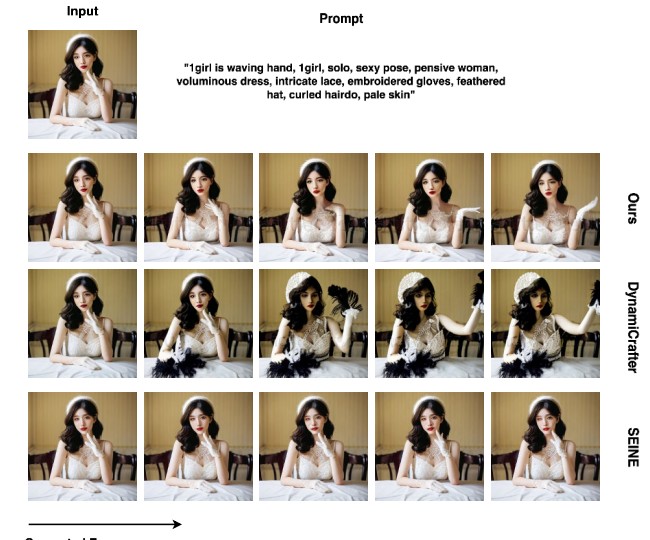

*Figure 5.* **Qualitative comparisons between FrameBridge and other baselines.** FrameBridge outperforms diffusion baseline methods in appearance consistency and video quality.

only change the noisy latent as input-reparameterization of bridge models. The forward and backward process is still a bridge process and is different from diffusion I2V models.

**Reparameterization of Bridge Process.** In bridge process, the perturbed latent $\mathbf{z}_t$ at timestep $t$ can be written as the linear combination of $\mathbf{z}_0$, $\mathbf{z}_T$ and a Gaussian noise $\boldsymbol{\epsilon}$: $\mathbf{z}_t = a_t\mathbf{z}_0 + b_t\mathbf{z}_T + c_t\boldsymbol{\epsilon}$ (detailed expression of $a_t, b_t, c_t$ can be found in Equation (12)), which takes a different form from $\alpha_t\mathbf{z}_0 + \sigma_t\boldsymbol{\epsilon}$ in diffusion models. Therefore, the pre-trained diffusion models have limited ability to directly denoise such a $\mathbf{z}_t$, which impairs effective fine-tuning. To match the distributions of $\mathbf{z}_t$, we reparameterize the bridge process by

$$\tilde{\mathbf{z}}_t = \frac{\mathbf{z}_t - b_t\mathbf{z}_T}{\sqrt{a_t^2 + c_t^2}} = \frac{a_t}{\sqrt{a_t^2 + c_t^2}}\mathbf{z}_0 + \frac{c_t}{\sqrt{a_t^2 + c_t^2}}\boldsymbol{\epsilon}. \quad (7)$$

Then, $\tilde{\mathbf{z}}_t$ can be represented as the combination of clean data $\mathbf{z}_0$ and a Gaussian noise, with the squre sum of coefficients equal to 1. Thus, the reparameterized bridge process $\tilde{\mathbf{z}}_t$ exactly aligns with a VP diffusion process.

**SNR-based Latent Alignment** Although the marginal distribution of $\tilde{\mathbf{z}}_t$ resembles that of a diffusion process, there is still a mismatch between the input of bridge models and pre-trained diffusion models (*i.e.*, $(\tilde{\mathbf{z}}_t, t)$ and $(\alpha_t\mathbf{z}_0 + \sigma_t\boldsymbol{\epsilon}, t)$), as it is not guaranteed that $\frac{a_t}{\sqrt{a_t^2 + c_t^2}} = \alpha_t$, $\frac{c_t}{\sqrt{a_t^2 + c_t^2}} = \sigma_t$ (see Figure 3). To handle that, we change the timestep $t$ to another $\tilde{t}$ such that $\alpha_{\tilde{t}} = \frac{a_t}{\sqrt{a_t^2 + c_t^2}}$, $\sigma_{\tilde{t}} = \frac{c_t}{\sqrt{a_t^2 + c_t^2}}$, and then $\tilde{\mathbf{z}}_t$ has the same SNR as $\alpha_{\tilde{t}}\mathbf{z}_0 + \sigma_{\tilde{t}}\boldsymbol{\epsilon}$ in diffusion process. According to the above derivation, we reparameterize the input of bridge models as

$\boldsymbol{\epsilon}_{\theta,bridge}^{\mathring{\Psi}}(\mathbf{z}_t, t, i, c) \triangleq \boldsymbol{\epsilon}_{\theta,aligned}^{\mathring{\Psi}}(\tilde{\mathbf{z}}_t, \tilde{t}, i, c)$, and initialize $\boldsymbol{\epsilon}_{\theta,aligned}^{\mathring{\Psi}}$ with the pre-trained T2V diffusion models. SAF enables bridge models to fully exploit the denoising capability of pre-trained diffusion models as the marginal distribution of $\tilde{\mathbf{z}}_t$ aligned with $\alpha_{\tilde{t}}\mathbf{z}_0 + \sigma_{\tilde{t}}\boldsymbol{\epsilon}$. We provide more details in Appendix A.

### 4.3 Improved Prior

By establishing a *data-to-data* process for I2V synthesis, we have been able to reduce the distance between the prior and the target from *noise-to-frames* to *frames-to-frames*, and therefore facilitate the generation process and aim at improving the synthesis quality. To further demonstrate the function of improving prior information for I2V synthesis, we extend our design of FrameBridge from replicated initial frame $\mathbf{z}^i$ to neural representations $F_\eta(z^i, c)$, which serves as a stronger prior for video frames.

As shown in Figure 4, although the static frame has provided indicative information such as the appearance details of the background and different objects, it may not be informative for the motion information in consecutive frames. When the distance between the prior frame and the target frame is large, bridge models are faced with the challenge to generate the motion trajectory. Therefore, we present a stronger prior than simply duplicating the initial frame, *neural prior*, which achieves a coarse estimation of the target at first, and then bridge models generate the high-quality target from this coarse estimation.

Considering bridge models synthesize target data with itera-

*Table 1.* Zero-shot I2V generation on UCF-101 and MSR-VTT ($256 \times 256$, 16 frames). w/o SAF means FrameBridge without SAF techniques when fine-tuning. For each metric, we mark the best one with † and the second one with ‡. Iterated videos is the number of videos iterated during the training of model (batch size × iterations). * : results reported in Xing et al. (2024). ** : reproduced with the open-sourced training code [2].

| Method | Iterated Videos | UCF-101 | | | MSR-VTT | | |
|---|---|---|---|---|---|---|---|
| | | FVD ↓ | IS ↑ | PIC ↑ | FVD ↓ | CLIPSIM ↑ | PIC ↑ |
| SVD (Blattmann et al., 2023) | – | 236 | – | – | 114 | – | – |
| SEINE (Chen et al., 2023b) | – | 461 | 22.32 | 0.6665 | 245 | **0.2250**† | 0.6848 |
| ConsistI2V (Ren et al.) | 32.64M | **202**† | 39.76 | **0.7638**† | 106 | 0.2249 | **0.7551**‡ |
| SparseCtrl (Guo et al., 2025) | – | 722 | 19.45 | 0.4818 | 311 | 0.2245 | 0.4382 |
| I2VGen-XL* (Zhang et al., 2023) | – | 571 | – | 0.5313 | 289 | – | 0.5352 |
| DynamiCrafter** (Xing et al., 2024) | 1.28M | 485 | 29.46 | 0.6266 | 192 | 0.2245 | 0.6131 |
| DynamiCrafter* | 6.4M | 429 | – | 0.6078 | 234 | – | 0.5803 |
| FrameBridge-VideoCrafter (w/o SAF) | 1.28M | 433 | 38.61 | 0.5989 | 229 | 0.2246 | 0.5559 |
| FrameBridge-VideoCrafter (w/ SAF) | 1.28M | 312 | **39.89**‡ | 0.6697 | 99 | **0.2250**† | 0.6963 |
| FrameBridge-VideoCrafter (w/ SAF) | 6.4M | 258 | **44.13**† | 0.7274 | **95**† | **0.2250**† | 0.7142 |
| FrameBridge-CogVideoX (w/ SAF) | 6.4M | **235**‡ | 39.83 | **0.7563**‡ | 96‡ | **0.2250**† | **0.7566**† |

*Table 2.* VBench-I2V (Huang et al., 2024a;b) scores for different I2V models. For each metric, we mark the best one with † and the second one with ‡ (higher score means better performance). The abbreviations represents Camera Motion (CM), I2V-Subject Consistency (I2V-SC), I2V-Background Consistency (I2V-BC), Subject Consistency (SC), Background Consistency (BC), Motion Smoothness (MS), Dynamic Degree (DD), Aesthetic Quality (AQ), Imaging Quality (IQ). Total Score: weighted average of all dimensions which evaluates the overall quality. Scores are calculated with the official code of VBench. * : results reported in Huang et al. (2024b).

| Model | Total Score | Detailed Qulity Dimensions | | | | | | | | |
|---|---|---|---|---|---|---|---|---|---|---|
| | | CM | I2V-SC | I2V-BC | SC | BC | MS | DD | AQ | IQ |
| DynamiCrafter-256 | 84.35 | 22.18 | 95.40 | 96.22 | 94.60 | 98.30 | **97.82**‡ | 38.69 | **59.40**† | 62.29 |
| SEINE-256 × 256 | 82.12 | 15.91 | 93.45 | 94.21 | 93.94 | 97.01 | 96.20 | 24.55 | 56.55 | **70.52**‡ |
| SEINE-512 × 320* | 83.49 | 23.36 | 94.85 | 94.02 | 94.20 | 97.26 | 96.68 | 34.31 | 58.42 | **70.97**† |
| SparseCtrl | 80.34 | 25.82 | 88.39 | 92.46 | 85.08 | 93.81 | 94.25 | **81.95**† | 49.88 | 69.35 |
| ConsistI2V* | 83.30 | **33.60**‡ | 94.69 | 94.57 | **95.27**† | 98.28 | 97.38 | 18.62 | 59.00 | 66.92 |
| FrameBridge-VideoCrafter | **85.37**‡ | 30.72 | **96.24**† | **97.25**† | 94.63‡ | **98.92**† | **98.51**† | 35.77 | 59.38‡ | 63.28 |
| FrameBridge-CogVideoX | **85.93**† | **92.06**† | 95.42‡ | 97.13‡ | 93.60 | **98.62**‡ | 97.57 | 48.29‡ | 54.28 | 60.00 |

tive sampling steps, we develop a one-step mapping-based prior network taking both image latent $z^i$ and text or label condition $c$ as input, and separately train the prior network with a regression loss in latent space:

$$\mathcal{L}_p(\eta) = \mathbb{E}_{(\mathbf{z}, z^i, c) \sim p_{data}(\mathbf{z}, z^i, c)} \left[ \left\| F_\eta(z^i, c) - \mathbf{z} \right\|^2 \right]. \quad (8)$$

With this objective, it can be proved that $F_\eta(z^i, c)$ learns to predict the mean value of subsequent frames, as shown in Appendix A. Given pre-trained $F_\eta(z^i, c)$, we build FrameBridge-NP from its output and target video latent $\mathbf{z}$ by replacing the prior $\mathbf{z}_T$ in Equation (6) with the neural prior $F_\eta(z^i, c)$. More details of the training and sampling algorithm can be found in Appendix B.

In generation, neural prior model $F_\eta(z^i, c)$ provide a coarse estimation with a single deterministic step, which is closer

to the target than the provided initial frame, and bridge model synthesize the video target with a coarse-to-fine iterative sampling process. Although more advanced methods can be designed to further improve neural prior, we present a design with simple training objective and one-step sampling, demonstrating the performance of enhancing prior information on I2V synthesis.

## 5 Experiments

We carry out experiments on UCF-101 (Soomro, 2012) and WebVid-2M (Bain et al., 2021) datasets to demonstrate the advantages of our data-to-data generation framework for I2V tasks. More details can be found in Appendix D.

---

[2]https://github.com/Doubiiu/DynamiCrafter

*Table 3.* Non-zero-shot I2V generation on UCF-101. The best and second results are marked with † and ‡.

| Method | FVD ↓ | IS ↑ | PIC ↑ |
|---|---|---|---|
| ExtDM | 649 | 21.37 | – |
| VDT-I2V | 171 | 62.61 | 0.7401 |
| FrameBridge | **154**‡ | **64.01**† | **0.7443**‡ |
| FrameBridge-NP | **122**† | 63.60‡ | **0.7662**† |

*Table 4.* Ablation of SAF technique on UCF-101 (non-zero-shot).

| Method | Iterations | FVD ↓ | IS ↑ | PIC ↑ |
|---|---|---|---|---|
| Diffusion | 10k | 176 | 53.60 | 0.7011 |
| Bridge (w/o SAF) | 10k | 176 | 53.93 | 0.7371 |
| Bridge (w/o SAF) | 5k | 284 | 49.40 | 0.6557 |
| Bridge (w/ SAF) | 5k | **141** | **55.98** | **0.8200** |

## 5.1 Fine-tuning from pre-trained diffusion models

Following Xing et al. (2024), we fine-tune text-conditional FrameBridge model with replicated prior $z^i$ from the open-sourced T2V diffusion model VideoCrafter1 (Chen et al., 2023a) and CogVideoX-2B (Yang et al., 2024b) on WebVid-2M dataset.

**Comparison with Baselines** We choose DynamiCrafter (Xing et al., 2024), SEINE (Chen et al., 2023b), I2VGen-XL (Zhang et al., 2023), SVD (Blattmann et al., 2023), ConsistI2V (Ren et al.) and SparseCtrl (Guo et al., 2025) as text-conditional I2V baselines. Table 1 shows zero-shot metrics on UCF-101 and MSR-VTT after fine-tuning on WebVid-2M. *Note that DynamiCrafter trained with 6.4M videos is a direct counterpart of FrameBridge-VideoCrafter, which uses the same model architecture, base T2V diffusion model and training budget*, which shows that powerful I2V models can achieve better generation performance by replacing diffusion process with a *data-to-data* bridge process. We also evaluate FrameBridge and other baselines with a comprehensive benchmark for video quality, *i.e.*, VBench-I2V (Huang et al., 2024a;b) (see Table 2, all the scores are calculated with the official code[3]). FrameBridge can effectively leverage the knowledge from pre-trained T2V diffusion models and generate videos with higher quality and consistency than the diffusion counterparts. We further discuss the trade-off between the dynamic degree and consistency in Appendix C.1. Qualitative results are shown in Figure 5. In the Figure, both FrameBridge and DynamiCrafter model are fine-tuned from VideoCrafter1 with 20k steps. To the best of our knowledge, our trial is the first time to fine-tune bridge models from pre-trained diffusion models.

---

[3]https://github.com/Vchitect/VBench

*Table 5.* Ablation of neural prior. Condition means whether the model conditions on $F_\eta(z^i, c)$.

| Method | Prior | Condition | FVD ↓ |
|---|---|---|---|
| VDT-I2V | Gaussian | ✗ | 171 |
| VDT-I2V | Gaussian | ✓ | 132 |
| FrameBridge | replicated | ✗ | 154 |
| FrameBridge | replicated | ✓ | 129 |
| FrameBridge-NP | neural | ✓ | **122** |

## 5.2 Neural Prior for Bridge Models

We train class-conditional FrameBridge model with neural prior (FrameBridge-NP) on UCF-101 based on the model of Latte-S/2 (Ma et al., 2024b) by replacing diffusion process with the Bridge-gmax bridge process (Chen et al., 2023c).

**Comparison with Baselines** We reproduce two diffusion models ExtDM (Zhang et al., 2024b) and VDT (Lu et al.) on UCF-101 dataset for the class-conditional I2V task as our baselines. Table 3 shows that FrameBridge-NP has superior video quality and consistency with condition images. Here VDT-I2V is a direct counterpart of FrameBridge models as they share the same network architecture and training configurations. More qualitative results are shown in Appendix F. The experiments reveal that bridge-based I2V models outperform their diffusion counterparts with both replicated prior and neural prior, justifying the usage of the *data-to-data* generation process for I2V tasks. Additionally, FrameBridge can further benefit from neural prior $F_\eta(z^i, c)$ as it actually narrows the gap between the prior and data distribution of bridge process.

## 5.3 Ablation Studies

**SNR-Aligned Fine-tuning** When fine-tuned with SAF, FrameBridge can leverage the pre-trained T2V diffusion models efficiently and effectively. To ablate on the SAF technique, we fine-tune a pre-trained class-conditional video generation model Latte-XL/2 on UCF-101. Table 4 shows that SAF improves fine-tuning performance of FrameBridge. To conduct an ablation under the WebVid-2M training setting, we also fine-tune FrameBridge models from VideoCrafter1 and CogVideoX-2B with the same configuration except the usage of SAF technique, and compare the zero-shot metrics in Appendix C.5.

**Neural Prior** To showcase the effectiveness of neural prior, we compare five different models varying in priors and network conditions. More details of the configurations can be found in Appendix D. Results in Table 5 reveal that $F_\eta(z^i, c)$ is indeed more informative than a single frame $z^i$ and can be fully utilized by FrameBridge through the

change of prior.

## 6 Conclusions

In this work, we propose FrameBridge, building a *data-to-data* generation process, which matches the *frame-to-frames* nature of this task, and therefore further improving the I2V synthesis quality of strong diffusion baselines. Additionally, targeting at two typical scenarios of training I2V models, namely fine-tuning from pre-trained diffusion models and training from scratch, we present SNR-Aligned Fine-tuning (SAF) and neural prior respectively to further improve the generation quality of FrameBridge. Extensive experiments show that FrameBridge generate videos with enhanced appearance consistency with image condition and improved temporal coherence, demonstrating the advantages of Frame-Bridge and the effectiveness of two proposed techniques.

## Acknowledgement

The authors sincerely thank Min Zhao, Chendong Xiang, Kaiwen Zheng for the insightful discussions and suggestions. This work was supported by the NSFC Projects (Nos. 62376131, 92270001, U24A20342). J.Z is also supported by the XPlorer Prize.

## Impact Statement

Our method FrameBridge can improve the quality of image-to-video generation, and the proposed techniques, *i.e.* SAF and neural prior, can furthur enhance FrameBridge. However, our method is broadly applicable, and FrameBridge I2V models can be fine-tuned from various T2V diffusion models, which potentially leads to the misuse of I2V generation models.

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

# A  Proof and Derivation

## A.1  Basics of Denoising Diffusion Bridge Model (DDBM)

We provide the derivations of $p_{t,bridge}(\mathbf{z}_t|\mathbf{z}_0, \mathbf{z}_T, z^i, c)$ and $h(\mathbf{z}, t, \mathbf{y}, z^i, c)$ used in Section 4.1.

Similar to the proofs in (Zhou et al.), we calculate $p_{t,bridge}(\mathbf{z}_t|\mathbf{z}_0, \mathbf{z}_T, z^i, c)$ by applying Bayes' rule:

$$
\begin{aligned}
p_{t,bridge}(\mathbf{z}_t|\mathbf{z}_0, \mathbf{z}_T, z^i, c) = p_{t,diff}(\mathbf{z}_t|\mathbf{z}_0, \mathbf{z}_T, z^i, c) &= \frac{p_{T,diff}(\mathbf{z}_T|\mathbf{z}_t, \mathbf{z}_0, z^i, c) p_{t,diff}(\mathbf{z}_t|\mathbf{z}_0, z^i, c)}{p_{t,diff}(\mathbf{z}_T|\mathbf{z}_0, z^i, c)} \\
&\overset{①}{=} \frac{p_{T,diff}(\mathbf{z}_T|\mathbf{z}_t) p_{t,diff}(\mathbf{z}_t|\mathbf{z}_0)}{p_{T,diff}(\mathbf{z}_T|\mathbf{z}_0)}.
\end{aligned}
\tag{9}
$$

① uses the Markovian of the diffusion process $\mathbf{z}_t$ (Kingma et al., 2021).

The perturbation kernels $p_{T,diff}(\mathbf{z}_T|\mathbf{z}_t), p_{t,diff}(\mathbf{z}_t|\mathbf{z}_0), p_{T,diff}(\mathbf{z}_T|\mathbf{z}_0)$ is Gaussian and takes the form of:

$$
\begin{aligned}
p_{T,diff}(\mathbf{z}_T|\mathbf{z}_t) &= \mathcal{N}(\mathbf{z}_T; \frac{\alpha_T}{\alpha_t}\mathbf{z}_t, (\sigma_T^2 - \frac{\alpha_T^2}{\alpha_t^2}\sigma_t^2)I), \\
p_{t,diff}(\mathbf{z}_t|\mathbf{z}_0) &= \mathcal{N}(\mathbf{z}_t; \alpha_t\mathbf{z}_0, \sigma_t^2 I), \\
p_{T,diff}(\mathbf{z}_T|\mathbf{z}_0) &= \mathcal{N}(\mathbf{z}_T; \alpha_T\mathbf{z}_0, \sigma_T^2 I).
\end{aligned}
\tag{10}
$$

Following (Zhou et al.), it can be derived that $p_{t,bridge}(\mathbf{z}_t|\mathbf{z}_T, \mathbf{z}_0, z^i, c)$ is also Gaussian, and $p_{t,bridge}(\mathbf{z}_t|\mathbf{z}_T, \mathbf{z}_0, z^i, c) = \mathcal{N}(\mathbf{z}_t; \mu_t(\mathbf{z}_0, \mathbf{z}_T), \sigma_{t,bridge}^2 I)$, where

$$
\begin{aligned}
\mu_t(\mathbf{z}_0, \mathbf{z}_T) &= \alpha_t(1 - \frac{\text{SNR}_T}{\text{SNR}_t})\mathbf{z}_0 + \frac{\text{SNR}_T}{\text{SNR}_t}\frac{\alpha_t}{\alpha_T}\mathbf{z}_T, \\
\sigma_{t,bridge}^2 &= \sigma_t^2(1 - \frac{\text{SNR}_T}{\text{SNR}_t}).
\end{aligned}
\tag{11}
$$

Specifically, $\mathbf{z}_t$ of bridge process can be reparameterized by $\mathbf{z}_t = a_t\mathbf{z}_0 + b_t\mathbf{z}_T + c_t\boldsymbol{\epsilon}$, where

$$
\begin{aligned}
a_t &= \alpha_t(1 - \frac{\text{SNR}_T}{\text{SNR}_t}), \\
b_t &= \frac{\text{SNR}_T}{\text{SNR}_t}\frac{\alpha_t}{\alpha_T}, \\
c_t &= \sqrt{\sigma_t^2(1 - \frac{\text{SNR}_T}{\text{SNR}_t})}.
\end{aligned}
\tag{12}
$$

Here, $\text{SNR}_t = \frac{\alpha_t^2}{\sigma_t^2}$ (Kingma et al., 2021) is the signal-to-noise ratio of diffusion process.

Then we calculate $h(\mathbf{z}, t, \mathbf{y}, z^i, c) = \nabla_{\mathbf{z}_t} \log p_{T,diff}(\mathbf{z}_T|\mathbf{z}_t)|_{\mathbf{z}_t=\mathbf{z}, \mathbf{z}_T=\mathbf{y}}$.

As $p_{T,diff}(\mathbf{z}_T|\mathbf{z}_t) = \mathcal{N}(\mathbf{z}_T; \frac{\alpha_T}{\alpha_t}\mathbf{z}_t, (\sigma_T^2 - \frac{\alpha_T^2}{\alpha_t^2}\sigma_t^2)I)$, we have

$$
p_{T,diff}(\mathbf{z}_T|\mathbf{z}_t) = \frac{1}{\sqrt{2\pi(\sigma_T^2 - \frac{\alpha_T^2}{\alpha_t^2}\sigma_t^2)}^D} \exp\left(-\frac{\left\|\mathbf{z}_T - \frac{\alpha_T}{\alpha_t}\mathbf{z}_t\right\|^2}{2(\sigma_T^2 - \frac{\alpha_T^2}{\alpha_t^2}\sigma_t^2)}\right),
\tag{13}
$$

$$
\log p_{T,diff}(\mathbf{z}_T|\mathbf{z}_t) = -\frac{\left\|\mathbf{z}_T - \frac{\alpha_T}{\alpha_t}\mathbf{z}_t\right\|^2}{2(\sigma_T^2 - \frac{\alpha_T^2}{\alpha_t^2}\sigma_t^2)} + C,
\tag{14}
$$

where $C$ is a constant independent of $\mathbf{z}_T$.

$$\nabla_{\mathbf{z}_t} \log p_{T,diff}(\mathbf{z}_T|\mathbf{z}_t) = \nabla_{\mathbf{z}_t}\left(-\frac{\left\|\mathbf{z}_T - \frac{\alpha_T}{\alpha_t}\mathbf{z}_t\right\|^2}{2(\sigma_T^2 - \frac{\alpha_T^2}{\alpha_t^2}\sigma_t^2)}\right) = -\frac{\mathbf{z}_T - \frac{\alpha_T}{\alpha_t}\mathbf{z}_t}{(\sigma_T^2 - \frac{\alpha_T^2}{\alpha_t^2}\sigma_t^2)}. \tag{15}$$

So, $\boldsymbol{h}(\mathbf{z}, t, \mathbf{y}, z^i, c) = -\frac{\mathbf{y} - \frac{\alpha_T}{\alpha_t}\mathbf{z}}{(\sigma_T^2 - \frac{\alpha_T^2}{\alpha_t^2}\sigma_t^2)}$. Note that for the diffusion process we commonly use, $\frac{\alpha_T}{\alpha_t} \approx 0$ and $\sigma_T \approx 1$, and we have $\boldsymbol{h}(\mathbf{z}, t, \mathbf{y}, z^i, c) \approx -\mathbf{y}$.

## A.2 Parameterization of FrameBridge

**Proposition 1.** *The score estimation $\boldsymbol{s}_\theta(\mathbf{z}_t, t, \mathbf{z}_T, z^i, c)$ of bridge process $p_{t,bridge}(\mathbf{z}_t|\mathbf{z}_T, z^i, c)$ can be reparamterized by*

$$\boldsymbol{s}_\theta(\mathbf{z}_t, t, \mathbf{z}_T, z^i, c) = -\frac{1}{\sigma_t}\boldsymbol{\epsilon}_\theta^{\hat{\Psi}}(\mathbf{z}_t, t, \mathbf{z}_T, z^i, c) - \frac{\mathrm{SNR}_T}{\mathrm{SNR}_t}\frac{\mathbf{z}_t - \frac{\alpha_t}{\alpha_T}\mathbf{z}_T}{\sigma_t^2(1 - \frac{\mathrm{SNR}_T}{\mathrm{SNR}_t})}, \tag{16}$$

*where $\mathrm{SNR}_t = \frac{\alpha_t^2}{\sigma_t^2}$, and $\boldsymbol{\epsilon}_\theta^{\hat{\Psi}}(\mathbf{z}_t, t, \mathbf{z}_T, z^i, c)$ is trained with the objective*

$$\mathcal{L}_{bridge}(\theta) = \mathbb{E}_{\substack{(\mathbf{z}_0, z^i, c) \sim p_{data}(\mathbf{z}_0, z^i, c), \\ \mathbf{z}_T = \mathbf{z}^i, t, \mathbf{z}_t \sim p_{t,bridge}(\mathbf{z}_t|\mathbf{z}_0, \mathbf{z}_T, z^i, c)}} \left[\tilde{\lambda}(t)\left\|\boldsymbol{\epsilon}_\theta^{\hat{\Psi}}(\mathbf{z}_t, t, \mathbf{z}_T, z^i, c) - \frac{\mathbf{z}_t - \alpha_t\mathbf{z}_0}{\sigma_t}\right\|^2\right]. \tag{17}$$

*Here $\tilde{\lambda}(t)$ is the weight function of timestep $t$ and we take $\tilde{\lambda}(t) = 1$ unless otherwise specified.*

*When $\mathrm{SNR}_T \approx 0$(which is often the case for diffusion process), there exists $\epsilon$ such that*

$$\boldsymbol{s}_\theta(\mathbf{z}_t, t, \mathbf{z}_T, z^i, c) \approx -\frac{1}{\sigma_t}\boldsymbol{\epsilon}_\theta^{\hat{\Psi}}(\mathbf{z}_t, t, \mathbf{z}_T, z^i, c), \quad \forall t \in [\epsilon, T - \epsilon]. \tag{18}$$

*Proof.* We denote the desnoising target $\frac{\mathbf{z}_t - \alpha_t\mathbf{z}_0}{\sigma_t}$ by $\boldsymbol{\epsilon}^{\hat{\Psi}}(\mathbf{z}_t, \mathbf{z}_0, t)$, and define $a_t = \alpha_t(1 - \frac{\mathrm{SNR}_T}{\mathrm{SNR}_t})$, $b_t = \frac{\mathrm{SNR}_T}{\mathrm{SNR}_t}\frac{\alpha_t}{\alpha_T}$, $c_t = \sqrt{\sigma_t^2(1 - \frac{\mathrm{SNR}_T}{\mathrm{SNR}_t})}$.

From Equation (11), we have

$$\nabla_{\mathbf{z}} \log p_{t,bridge}(\mathbf{z}|\mathbf{z}_0, \mathbf{z}_T)|_{\mathbf{z}=\mathbf{z}_t, \mathbf{z}_T=\mathbf{z}^i} = -\frac{\mathbf{z}_t - a_t\mathbf{z}_0 - b_t\mathbf{z}^i}{c_t^2}, \tag{19}$$

which is the target of Denoising Bridge Score Matching (Zhou et al.). Our goal is to represent this target with $\mathbf{z}_t$, $\mathbf{z}_T$, and $\boldsymbol{\epsilon}^{\hat{\Psi}}(\mathbf{z}_t, \mathbf{z}_0, t)$.

From the definition of $\boldsymbol{\epsilon}^{\hat{\Psi}}(\mathbf{z}_t, \mathbf{z}_0, t)$, we have

$$\mathbf{z}_0 = \frac{\mathbf{z}_t - \sigma_t\boldsymbol{\epsilon}^{\hat{\Psi}}(\mathbf{z}_t, \mathbf{z}_0, t)}{\alpha_t}. \tag{20}$$

Plug it into Equation (19), it can be derived that

$$
\begin{aligned}
\nabla_{\mathbf{z}} \log p_{t,bridge}(\mathbf{z}|\mathbf{z}_0, \mathbf{z}_T)|_{\mathbf{z}=\mathbf{z}_t, \mathbf{z}_T=\mathbf{z}^i} &= -\frac{\mathbf{z}_t - a_t \frac{\mathbf{z}_t - \sigma_t \boldsymbol{\epsilon}^{\hat{\Psi}}(\mathbf{z}_t, \mathbf{z}_0, t)}{\alpha_t} - b_t \mathbf{z}^i}{c_t^2} \\
&= -\frac{\alpha_t \mathbf{z}_t - a_t \mathbf{z}_t + a_t \sigma_t \boldsymbol{\epsilon}^{\hat{\Psi}}(\mathbf{z}_t, \mathbf{z}_0, t) - \alpha_t b_t \mathbf{z}_T}{\alpha_t c_t^2} \\
&= -\frac{a_t \sigma_t \boldsymbol{\epsilon}^{\hat{\Psi}}(\mathbf{z}_t, \mathbf{z}_0, t)}{\alpha_t c_t^2} - \frac{(\alpha_t - a_t)\mathbf{z}_t - \alpha_t b_t \mathbf{z}^i}{\alpha_t c_t^2} \\
&= -\frac{1}{\sigma_t}\boldsymbol{\epsilon}^{\hat{\Psi}}(\mathbf{z}_t, \mathbf{z}_0, t) - \frac{\alpha_t \frac{\text{SNR}_T}{\text{SNR}_t}\mathbf{z}_t - \frac{\alpha_t^2}{\alpha_T}\frac{\text{SNR}_T}{\text{SNR}_t}\mathbf{z}^i}{\alpha_t \sigma_t^2 (1 - \frac{\text{SNR}_T}{\text{SNR}_t})} \\
&= -\frac{1}{\sigma_t}\boldsymbol{\epsilon}^{\hat{\Psi}}(\mathbf{z}_t, \mathbf{z}_0, t) - \frac{\text{SNR}_T}{\text{SNR}_t}\frac{\mathbf{z}_t - \frac{\alpha_t}{\alpha_T}\mathbf{z}^i}{\sigma_t^2 (1 - \frac{\text{SNR}_T}{\text{SNR}_t})},
\end{aligned}
\tag{21}
$$

As the Denoising Bridge Score Matching takes the form of

$$
\mathcal{L}_{bridge}(\theta) = \mathbb{E}_{(\mathbf{z}_0, z^i, c).\mathbf{z}_T=\mathbf{z}^i, t, \mathbf{z}_t}\left[\lambda(t)\left\|\boldsymbol{s}_\theta(\mathbf{z}_t, t, \mathbf{z}_T, z^i, c) - \nabla_{\mathbf{z}}\log p_{t,bridge}(\mathbf{z}|\mathbf{z}_0, \mathbf{z}_T)|_{\mathbf{z}=\mathbf{z}_t, \mathbf{z}_T=\mathbf{z}^i}\right\|^2\right],
\tag{22}
$$

when we parameterize $\boldsymbol{s}_\theta(\mathbf{z}_t, t, \mathbf{z}_T, z^i, c) = -\frac{1}{\sigma_t}\boldsymbol{\epsilon}_\theta^{\hat{\Psi}}(\mathbf{z}_t, t, \mathbf{z}_T, z^i, c) - \frac{\text{SNR}_T}{\text{SNR}_t}\frac{\mathbf{z}_t - \frac{\alpha_t}{\alpha_T}\mathbf{z}_T}{\sigma_t^2(1 - \frac{\text{SNR}_T}{\text{SNR}_t})}$, the training objective can be written as

$$
\mathcal{L}_{bridge}(\theta) = \mathbb{E}_{(\mathbf{z}_0, z^i, c).\mathbf{z}_T=\mathbf{z}^i, t, \mathbf{z}_t}\left[\frac{\lambda(t)}{\sigma_t^2}\left\|\boldsymbol{\epsilon}_\theta^{\hat{\Psi}}(\mathbf{z}_t, t, \mathbf{z}_T, z^i, c) - \boldsymbol{\epsilon}^{\hat{\Psi}}(\mathbf{z}_t, \mathbf{z}_0, t)\right\|^2\right],
\tag{23}
$$

which proves the first part of the proposition if we take $\tilde{\lambda}(t) = \frac{\lambda(t)}{\sigma_t^2}$.

For the second part, when $\text{SNR}_T \approx 0$, there exists an $\epsilon > 0$, such that $\frac{1}{\sigma_t^2(1 - \frac{\text{SNR}_T}{\text{SNR}_t})}$ has an upper bound $M$. Since $\frac{\text{SNR}_T}{\text{SNR}_t}\frac{\alpha_t}{\alpha_T} = \alpha_T \frac{\sigma_t^2}{\alpha_t \sigma_T^2} \approx 0$ when $\text{SNR}_T \approx 0$, it can be directly inferenced from Equation (16) that $\boldsymbol{s}_\theta(\mathbf{z}_t, t, \mathbf{z}_T, z^i, c) \approx -\frac{1}{\sigma_t}\boldsymbol{\epsilon}_\theta^{\hat{\Psi}}(\mathbf{z}_t, t, \mathbf{z}_T, z^i, c)$. $\square$

*Remark* A.1. From the first part of the proposition, we parameterize bridge models to predict $\frac{\mathbf{z}_t - \alpha_t \mathbf{z}_0}{\sigma_t}$. It is similar to that used in Chen et al. (2023c) although their parameterization is derived from the forward-backward diffusion process of Schrödinger Bridge problems. The statement and proof of this proposition reveals that DDBM and Diffusion Schrödinger Bridges are closely related. Additionally, the second part shows that our parameterization resembles the Denoising Score Matching in diffusion models.

## A.3  SNR-Aligned Fine-tuning

**Existence and Uniqueness of $\tilde{t}$**  In Section 4.2, we need to find a $\tilde{t}$ such that $\alpha_{\tilde{t}} = \frac{a_t}{\sqrt{a_t^2 + c_t^2}}$, $\sigma_{\tilde{t}} = \frac{c_t}{\sqrt{a_t^2 + c_t^2}}$. Since $\frac{a_t^2}{c_t^2} = \frac{\alpha_t^2}{\sigma_t^2}(1 - \frac{\text{SNR}_T}{\text{SNR}_t}) = \text{SNR}_t - \text{SNR}_T$, it is a monotonically decreasing function of $t$. As $\text{SNR}_t$ is also a monotonically decreasing function which ranges over $(0, \infty)$, we can take $\tilde{t} = \text{SNR}^{-1}(\frac{a_t^2}{c_t^2})$ and the uniqueness of such $\tilde{t}$ can also be guaranteed. Next, we provide a more general form of SAF, where the schedule $\{\alpha_t, \sigma_t\}_{t\in[0,T]}$ of the pre-trained diffusion models and bridge models are not necessarily the same.

**Proposition 2.** *Suppose we fine-tune a Gaussian diffusion model $\tilde{\boldsymbol{\epsilon}}_\eta(\mathbf{z}_t, t, c)$ with schedule $\{\tilde{\alpha}_t, \tilde{\sigma}_t\}_{t\in[0,T]}$ to a diffusion bridge model $\boldsymbol{\epsilon}_{\theta,bridge}^{\hat{\Psi}}(\mathbf{z}_t, t, \mathbf{z}_T, z^i, c) \triangleq \boldsymbol{\epsilon}_{\theta,align}^{\hat{\Psi}}(\tilde{\mathbf{z}}_t, \tilde{t}, \mathbf{z}_T, z^i, c)$ with schedule $\{\alpha_t, \sigma_t\}_{t\in[0,T]}$. If we use the same dataset $p_{data}(\mathbf{z}_0, z^i, c)$ for training $\tilde{\boldsymbol{\epsilon}}_\eta(\mathbf{z}_t, t, c)$ and fine-tuning $\boldsymbol{\epsilon}_{\theta,align}^{\hat{\Psi}}(\tilde{\mathbf{z}}_t, \tilde{t}, \mathbf{z}_T, z^i, c)$. Then, for each c, the input $(\mathbf{z}_t, t)$ of $\tilde{\boldsymbol{\epsilon}}_\eta$ has the same marginal distribution as the input $(\tilde{\mathbf{z}}_t, \tilde{t})$ of $\boldsymbol{\epsilon}_{\theta,align}^{\hat{\Psi}}(\tilde{\mathbf{z}}_t, \tilde{t}, \mathbf{z}_T, z^i, c)$. Here*

$$
\begin{aligned}
\tilde{\mathbf{z}}_t &= \frac{\mathbf{z}_t - b_t \mathbf{z}^i}{\sqrt{a_t^2 + c_t^2}}, \\
\tilde{t} &= \widetilde{SNR}^{-1}(\frac{a_t^2}{c_t^2}).
\end{aligned}
\tag{24}
$$

$(\widetilde{SNR} = \frac{\tilde{\alpha}_t^2}{\tilde{\sigma}_t^2}$ is the signal-to-noise ratio of pre-trained diffusion models.)

*Proof.* Since $\widetilde{SNR}$ is also a monotonically decreasing function ranging over $(0, \infty)$, the uniqueness and existence of $\tilde{t}$ can also be guaranteed by the above analysis.

For a fixed $c, t$, we denote the probability density function of $\tilde{\mathbf{z}}_t$ by $q(\tilde{\mathbf{z}}_t; t)$. Then

$$
\begin{aligned}
q(\tilde{\mathbf{z}}_t; t) &= \int_{z^i} q(\tilde{\mathbf{z}}_t | z^i; t) p_{data}(z^i) \mathrm{d}z^i \\
&= \int_{z^i} \int_{\mathbf{z}_0} q(\tilde{\mathbf{z}}_t | \mathbf{z}_0, z^i; t) p_{data}(\mathbf{z}_0, z^i) \mathrm{d}\mathbf{z}_0 \mathrm{d}z^i \\
&= \int_{z^i} \int_{\mathbf{z}_0} \mathcal{N}(\tilde{\mathbf{z}}_t; \frac{a_t}{\sqrt{a_t^2 + c_t^2}} \mathbf{z}_0, \frac{c_t}{\sqrt{a_t^2 + c_t^2}} I) p_{data}(\mathbf{z}_0, z^i) \mathrm{d}\mathbf{z}_0 \mathrm{d}z^i \\
&= \int_{z^i} \int_{\mathbf{z}_0} \mathcal{N}(\tilde{\mathbf{z}}_t; \tilde{\alpha}_{\tilde{t}} \mathbf{z}_0, \tilde{\sigma}_{\tilde{t}}^2 I) p_{data}(\mathbf{z}_0, z^i) \mathrm{d}\mathbf{z}_0 \mathrm{d}z^i \\
&= \int_{\mathbf{z}_0} \mathcal{N}(\tilde{\mathbf{z}}_t; \tilde{\alpha}_{\tilde{t}} \mathbf{z}_0, \tilde{\sigma}_{\tilde{t}}^2 I) (\int_{z^i} p_{data}(\mathbf{z}_0, z^i) \mathrm{d}z^i) \mathrm{d}\mathbf{z}_0 \\
&= \int_{\mathbf{z}_0} \mathcal{N}(\tilde{\mathbf{z}}_t; \tilde{\alpha}_{\tilde{t}} \mathbf{z}_0, \tilde{\sigma}_{\tilde{t}}^2 I) p_{data}(\mathbf{z}_0) \mathrm{d}\mathbf{z}_0,
\end{aligned}
\tag{25}
$$

which equals to the marginal distribution of the pre-trained diffusion process $p_{t,diff}(\mathbf{z}_t)$.

$\square$

**Output Parameterization**  Our previous descriptions show how to align the input of the network when fine-tuning from T2V diffusion models to I2V bridge models. When the output parameterization of teacher diffusion models deviates significantly from the bridge parameterization $\epsilon^{\hat{\Psi}}$, we can also reparameterize the network output to achieve better alignment. We take CogVideoX-2B as an example, where v-prediction is used for teacher diffusion models. The teacher diffusion models predict $\alpha_{\tilde{t}} \epsilon - \sigma_{\tilde{t}} \mathbf{z}_0$ from $(\alpha_{\tilde{t}} \mathbf{z}_0 + \sigma_{\tilde{t}} \epsilon, \tilde{t})$. After the input alignment of bridge schedule, we have

$$
\begin{aligned}
\tilde{\mathbf{z}}_t &= \frac{a_t}{\sqrt{a_t^2 + c_t^2}} \mathbf{z}_0 + \frac{c_t}{\sqrt{a_t^2 + c_t^2}} \epsilon, \\
\alpha_{\tilde{t}} &= \frac{a_t}{\sqrt{a_t^2 + c_t^2}}, \quad \sigma_{\tilde{t}} = \frac{c_t}{\sqrt{a_t^2 + c_t^2}}.
\end{aligned}
\tag{26}
$$

To align the network output with the teacher, we can set the target of prediction as $\frac{a_t}{\sqrt{a_t^2 + c_t^2}} \epsilon - \frac{c_t}{\sqrt{a_t^2 + c_t^2}} \mathbf{z}_0$.

## A.4   Neural Prior with Regression Training Objective.

**Proposition 3.** *If we train $F_\eta(z^i, c)$ with the regression training objective*

$$
\mathcal{L}_p(\eta) = \mathbb{E}_{(\mathbf{z}_0, z^i, c) \sim p_{data}(\mathbf{z}_0, z^i, c)} \left[ \left\| F_\eta(z^i, c) - \mathbf{z}_0 \right\|^2 \right],
\tag{27}
$$

*and the neural network is optimized sufficiently, then we have*

$$
F_\eta(z^i, c) = F_\eta^*(z^i, c) \triangleq \mathbb{E}_{\mathbf{z}_0 \sim p_{data}(\mathbf{z}_0 | z^i, c)} [\mathbf{z}_0].
\tag{28}
$$

*Proof.* For each $(z^i, c)$, $\mathcal{L}_p(\eta)$ optimizes the following objective:

$$
\begin{aligned}
l_\eta(z^i, c) &= \mathbb{E}_{\mathbf{z}_0 \sim p_{data}(\mathbf{z}_0 | z^i, c)} \left[ \left\| F_\eta(z^i, c) - \mathbf{z}_0 \right\|^2 \right] \\
&= \left\| F_\eta(z^i, c) \right\|^2 - \langle F_\eta(z^i, c), \mathbb{E}_{\mathbf{z}_0 \sim p_{data}(\mathbf{z}_0 | z^i, c)} [\mathbf{z}_0] \rangle + \left\| \mathbb{E}_{\mathbf{z}_0 \sim p_{data}(\mathbf{z}_0 | z^i, c)} [\mathbf{z}_0] \right\|^2 \\
&= \left\| F_\eta(z^i, c) \right\|^2 - \langle F_\eta(z^i, c), \mathbb{E}_{\mathbf{z}_0 \sim p_{data}(\mathbf{z}_0 | z^i, c)} [\mathbf{z}_0] \rangle + C.
\end{aligned}
\tag{29}
$$

---

**Algorithm 1** Training algorithms for I2V diffusion models.

---

**Output:** Trained I2V diffusion model $\epsilon_\theta(\mathbf{z}_t, t, z^i, c)$.
Set diffusion process $\{\alpha_t, \sigma_t\}_{t=0}^T$.
**if** Fine-tuned from pre-trained diffuion model $\epsilon_\phi(\mathbf{z}_t, t, c)$ **then**
    Initialize $\epsilon_\theta$ with the weight of $\epsilon_\phi(\mathbf{z}_t, t, c)$.
**else**
    Randomly initialize $\epsilon_\theta(\mathbf{z}_t, t, z^i, c)$.
**end if**
**repeat**
    Sample data $(\mathbf{z}_0, c) \sim p_{data}(\mathbf{z}_0, c)$, timestep $t$ and $\mathbf{z}_t = \alpha_t \mathbf{z}_0 + \sigma_t \epsilon$, where $\epsilon \sim \mathcal{N}(0, I)$.
    Take the first frame of $\mathbf{z}_0$ as the image condition $z^i$.
    $l(\theta) = \left\| \epsilon_\theta(\mathbf{z}_t, t, z^i, c) - \epsilon \right\|^2$.
    Update $\theta$ with the optimizer and loss function $l(\theta)$
**until** Reach the training budget

---

**Algorithm 2** Sampling algorithms for FrameBridge.

---

**Output:** Video latent $\mathbf{z}_0$.
Prepare a trained FrameBridge model $\epsilon_\theta^{\hat{\Psi}}(\mathbf{z}_t, t, \mathbf{z}_T, z^i, c)$ and timestep schedule $0 = t_0 < t_1 < ... < t_N = T$.
Obtain the given input image $z^i$ and additional conditions $c$.
**if** Neural prior is used **then**
    $\mathbf{z}_T \leftarrow F_\eta(z^i, c)$. Here $F_\eta$ should be the same neural prior model used in the training process.)
**else**
    Construct $\mathbf{z}_T$ by replicating $z^i$.
**end if**
**for** $k = N$ downto 1 **do**
    Calculate the score function of bridge process $\nabla_\mathbf{z} \log p_{bridge, t_k}(\mathbf{z}|\mathbf{z}_T, z^i, c)|_{\mathbf{z}=\mathbf{z}_{t_k}}$ with $\epsilon_\theta^{\hat{\Psi}}(\mathbf{z}_{t_k}, t_k, \mathbf{z}_T, z^i, c)$.
    Utilize a SDE solver to solve the backward bridge SDE $d\mathbf{z}_t = \left[ \boldsymbol{f}(t)\mathbf{z}_t - g(t)^2 (\boldsymbol{s}(\mathbf{z}_t, t, \mathbf{z}_T, z^i, c) - \boldsymbol{h}(\mathbf{z}_t, t, \mathbf{z}_T, z^i, c)) \right] dt + g(t)d\bar{w}$ from $\mathbf{z}(t_k) = \mathbf{z}_{t_k}$ to obtain $\mathbf{z}_{t_{k-1}}$.
**end for**
Return $\mathbf{z}_0$.

---

where $C$ is a constant independent of $\eta$. When the network is optimized sufficiently, $l_\eta(z^i, c)$ takes the minimum for each $(z^i, c)$, so we have

$$F_\eta(z^i, c) = \arg\min_\mathbf{x} \left( \|\mathbf{x}\|^2 - \langle \mathbf{x}, \mathbb{E}_{\mathbf{z}_0 \sim p_{data}(\mathbf{z}_0|z^i, c)} [\mathbf{z}_0] \rangle \right) \tag{30}$$

It can be solved that $F_\eta(z^i, c) = \mathbb{E}_{\mathbf{z}_0 \sim p_{data}(\mathbf{z}_0|z^i, c)} [\mathbf{z}_0]$. $\qquad\square$

## B   Pseudo Code for the Training and Sampling of FrameBridge

We provide the pseudo code for the training and sampling process of FrameBridge (See Algorithm 4 and 2). Meanwhile, we also provide that of diffusion-based I2V models (See Algorithm 1 and 3) to show the distinctions between FrameBridge and diffusion-based I2V models.

## C   Detailed Analysis of I2V Generation Performance

In this section, we provide further discussions and analysis of the results provided in Section 5.

### C.1   Dynamic Degree of Generated Videos

As shown by (Zhao et al.), there is usually a trade-off between dynmaic motion and condition alignment for I2V models, and the high dynamic degree scores of some baseline models in Table 2 are at the cost of condition and temporal consistency. FrameBridge can reach a balance demonstrated by the multi-dimensional evaluation on VBench-I2V. Table 6 shows

---

**Algorithm 3** Sampling algorithms for I2V diffusion models.

---

    **Output:** Video latent $\mathbf{z}_0$.

    Prepare a trained I2V diffusion model $\epsilon_\theta(\mathbf{z}_t, t, z^i, c)$ and timestep schedule $0 = t_0 < t_1 < ... < t_N = T$.

    Obtain the given input image $z^i$ and additional conditions $c$.

    Sample a latent $\mathbf{z}_T \sim \mathcal{N}(0, \sigma_T^2 I)$.

    **for** $k = N$ downto 1 **do**

        Calculate the score function of diffusion process $\nabla_\mathbf{z} \log p_{diff,t_k}(\mathbf{z}|z^i, c)|_{\mathbf{z}=\mathbf{z}_{t_k}}$ with $\epsilon(\mathbf{z}_{t_k}, t_k, z^i, c)$.

        Utilize a SDE solver to solve the backward diffusion SDE $\mathrm{d}\mathbf{z}_t = \left[ \boldsymbol{f}(t)\mathbf{z}_t - g(t)^2 \nabla_{\mathbf{z}_t} \log p_{t,diff}(\mathbf{z}_t|z^i, c) \right] \mathrm{d}t + g(t)\mathrm{d}\bar{\mathbf{w}}$

        from $\mathbf{z}(t_k) = \mathbf{z}_{t_k}$ to obtain $\mathbf{z}_{t_{k-1}}$.

    **end for**

    Return $\mathbf{z}_0$.

---

*Table 6.* VBench-I2V scores related to the motion of videos for different I2V models. For all the evaluation dimensions, higher score means better performance. For results marked by *, we directly use the data of VBench-I2V Leaderboard.

| Model | Dynamic Degree | Temporal Flickering | Motion Smoothness |
|---|---|---|---|
| FrameBridge-VideoCrafter | 35.77 | **98.01** | **98.51** |
| DynamiCrafter-256 | **38.69** | 97.03 | 97.82 |
| SEINE-256 $\times$ 256 | 24.55 | 95.07 | 96.20 |
| SEINE-512 $\times$ 320* | 34.31 | 96.72 | 96.68 |
| SEINE-512 $\times$ 512* | 27.07 | 97.31 | 97.12 |
| ConsistI2V* | 18.62 | 97.56 | 97.38 |

VBench-I2V scores related to dynamic degree and temporal consistency.

Meanwhile, some techniques are proposed for I2V diffusion models to improve the dynamic degree and we find they are also applicable to FrameBridge. To be more specific, we fine-tune FrameBridge-VideoCrafter by adding noise to the image condition (Blattmann et al., 2023; Zhao et al.) and use higher value of frame-stride conditioning (Xing et al., 2024) respectively, and conduct a user study to evaluate the dynamic degree and overall video quality. We randomly sample 50 prompts from VBench-I2V and generate one video with each prompt for each model. Participants are asked two questions for each group of videos:

- Rank the videos according to the dynamic degree. Higher rank (i.e. lower ranking number) corresponds to higher dynamic degree.

- Rank the videos according to the overall quality. Higher rank (i.e. lower ranking number) corresponds to higher quality.

We recruited 18 participants and use Average User Ranking (AUR) as a preference metric (lower for better performance). The results are shown in Table 7.

## C.2 Content-Debiased FVD

Ge et al. (2024) points out that the FVD metric has a content bias and may misjudge the qualify of videos. As supplementary, we also provide the evaluation results of the Content-Debiased FVD (CD-FVD) on MSR-VTT in Table 8.

## C.3 Learning Curve of Video Quality

To illustrate the change of video quality during training, we reproduce the training process of DynamiCrafter for 20k iterations and compare the zero-shot CD-FVD metric on MSR-VTT dataset with a FrameBridge model trained during the training process. As we use the same training batch size and model structure for FrameBridge and DynamiCrafter in this experiment, the training budget for two models at the same training step is also the same. As demonstrated by Figure 6, the video quality of FrameBridge is superior to that of DynamiCrafter during the training process and it also converges faster than its diffusion counterpart (*i.e.*, DynamiCrafter).

*Table 7.* Results of user study. All the models are fine-tuned from VideoCrafter1. For FrameBridge-FrameStride, we increase the value of conditioning frame stride (from 3 to 5) when sampling. For FrameBridge-NoisyCondition, we add noise to the image condition in the fine-tuning process.

| Model | AUR of dynamic degree ↓ | AUR of overall quality ↓ |
|---|---|---|
| DynamiCrafter | 2.85 | 3.04 |
| FrameBridge | 2.74 | **2.26** |
| FrameBridge-FrameStride | **2.12** | 2.34 |
| FrameBridge-NoisyCondition | 2.29 | 2.35 |

*Table 8.* Zero-shot CD-FVD metric on MSR-VTT dataset. We also include the FVD metric as a reference

| Model | CD-FVD ↓ | FVD ↓ |
|---|---|---|
| DynamiCrafter | 207 | 234 |
| SEINE | 420 | 245 |
| ConsistI2V | 192 | 106 |
| SparseCtrl | 454 | 311 |
| FrameBridge-VideoCrafter | **148** | **95** |

### C.4 Sampling Efficiency of FrameBridge

Since sampling efficiency is also important for I2V models, we also conduct experiments to show the quality of videos sampled with different number of sampling timesteps and compare it with DynamiCrafter and SEINE. Figure C.4 shows that the quality of videos sampled by FrameBridge is better than that of DynamiCrafter and SEINE with different timesteps (*i.e.*, 250, 100, 50, 40, 20). Moreover, we also measure the actual execution time of the sampling algorithm and show the result in Figure C.4. As illustrated by these two figures, FrameBridge can achieve good balance between sample efficiency and video quality, and there is no significant degradation in video quality when decreasing the sampling timestep from 250 to 50 or even smaller.

### C.5 SNR-Aligned Fine-tuning on WebVid-2M

To ablate SAF technique on WebVid-2M, we fine-tune FrameBridge models from VideoCrafter1 with the same configuration except the usage of SAF for 1.6k steps. The zero-shot metrics are reported in Table 9. Similar ablation is conducted with FrameBridge models fine-tuned from CogVideoX-2B for 5k steps, and the zero-shot metrics are reported in Table 10.

## D   Experiment Details

We provide descriptions of the datasets and metrics used in our experiments, along with implementation details for different I2V models.

### D.1 Datasets

**UCF-101** is an open-sourced video dataset consisting of 13320 videos clips, and each video clip are categorized into one of the 101 action classes. There are three official train-test split, each of which divide the whole dataset into 9537 training video clips and 3783 test video clips. We use the whole dataset as the training data for I2V models trained from scratch on UCF-101, and use the test set to evaluate zero-shot metrics for models fine-tuned on WebVid-2M. When we evaluate zero-shot metrics on UCF-101 for text-conditional I2V models, we use the class label as the input text prompt.

**WebVid-2M** is an open-sourced dataset consisting of about 2.5 million video-text pairs, which is a subset of WebVid-10M. We only use WebVid-2M as the training data when fine-tuning I2V models from T2V diffusions in Section 5.1.

**MSR-VTT** is an open-sourced dataset consisting of 10000 video-text pairs, and we only use the test set to compute zero-shot metrics for fine-tuned models.

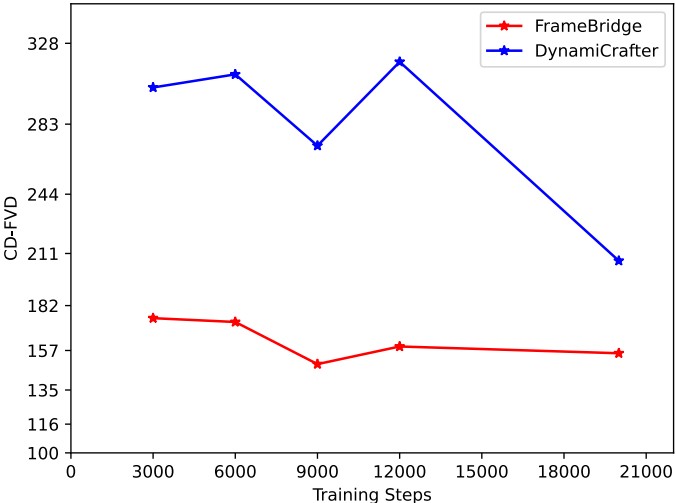

*Figure 6.* The learning curve of FrameBridge and DynamiCrafter.

*Table 9.* Zero-shot metrics on UCF-101 and MSR-VTT for FrameBridge-VideoCrafter models.

| Model | UCF-101 | | | MSR-VTT | | |
|---|---|---|---|---|---|---|
| | FVD ↓ | IS ↑ | PIC ↑ | FVD ↓ | CLIPSIM ↑ | PIC ↑ |
| FrameBridge-VideoCrafter (w/o SAF) | 431 | 45.88 | 0.6765 | 151 | 0.2248 | 0.6493 |
| FrameBridge-VideoCrafter (w/SAF) | 354 | 46.09 | 0.7060 | 132 | 0.2248 | 0.6778 |

**Preprocess of Training Data:** For both UCF-101 and WebVid-2M dataset, we sample 16 frames from each video clip with a fixed frame stride of 3 when training. Then we resize and center-crop the video clips to $256 \times 256$ before input it to the models.

### D.2 Metrics

**Fréchet Video Distance (** Unterthiner et al. (2018)**; FVD)** evaluates the quality of synthesized videos by computing the perceptual distance between videos sampled from the dataset and the models. We follow the protocol used in StyleGAN-V (Skorokhodov et al., 2022) to calculate FVD. First, we sample 2048 video clips with 16 frames and frame stride of 3 from the dataset. Then, we generate 2048 videos from the I2V models. All videos are resized to $256 \times 256$ before calculating FVD except for ExtDM. (ExtDM generate videos with resolution $64 \times 64$, so we compute FVD on this resolution.) After that, we extract features of those videos with the same I3D model used in the repository of StyleGAN-V [4] and calculate the Fréchet Distance.

**Inception Score (**Saito et al. (2017)**; IS)** also evaluates the quality of the generated videos. However, computing IS need a pre-trained classifier and we only apply this metric on UCF-101. When computing IS, we use the open-sourced evaluation code and pre-trained classifier for videos from the repository of StyleGAN-V.

**CLIPSIM (**Wu et al., 2021**)** evaluates the consistency between video frames and the text prompt by computing the average CLIP similarity score between each frame and the prompt. We use the VIT-B/32 CLIP model (Radford et al., 2021) when evaluating zero-shot metrics on MSR-VTT.

**PIC** is a metric used by Xing et al. (2024) to evaluate the consistency of video frames and the given image by the computing average Dreamsim (Fu et al., 2023) distance between generated frames and the image condition.

---

[4]https://github.com/universome/stylegan-v

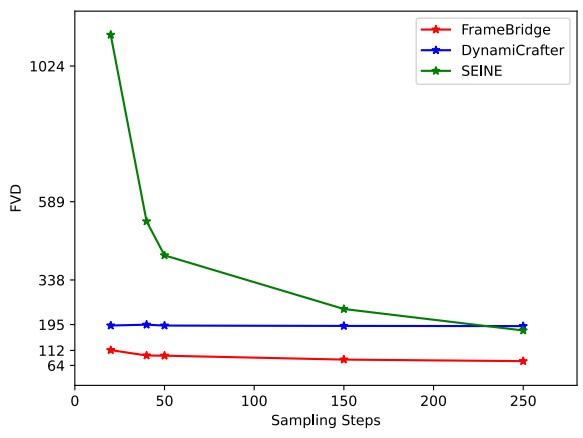
(a) Zero-shot FVD with different sampling timesteps

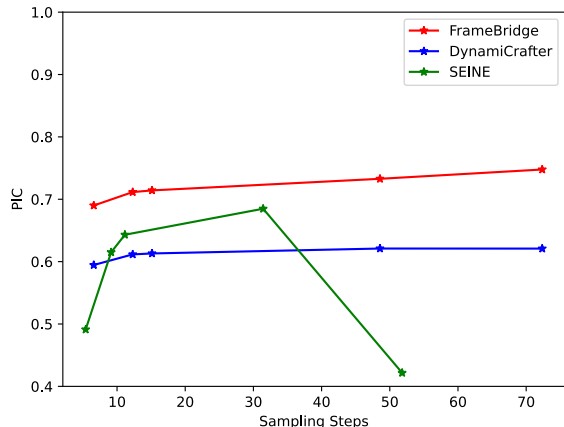
(b) Zero-shot PIC with different sampling timesteps

*Figure 7.* Video quality sampled with different number of timesteps.

*Table 10.* Zero-shot metrics on UCF-101 and MSR-VTT for FrameBridge-CogVideoX models.

| Model | UCF-101 | | | MSR-VTT | | |
|---|---|---|---|---|---|---|
| | FVD ↓ | IS ↑ | PIC ↑ | FVD ↓ | CLIPSIM ↑ | PIC ↑ |
| FrameBridge-CogVideoX (w/o SAF) | 359 | 36.84 | 0.5868 | 209 | 0.2250 | 0.6056 |
| FrameBridge-CogVideoX (w/SAF) | 347 | 41.12 | 0.6563 | 185 | 0.2250 | 0.6587 |

## D.3 Implementation of FrameBridge and Other Baselines

We offer the implementation details of I2V models which are fine-tuned on WebVid-2M or trained from scratch on UCF-101.

### D.3.1 FRAMEBRIDGE

**Fine-tuning on WebVid2M** For FrameBridge-VideoCrafter, we refer to the codebase of Dynamicrafter[5] to fine-tune FrameBridge, and initialize our model from the pre-trained VideoCrafter1 (Chen et al., 2023a) checkpoint. For FrameBridge-CogVideoX, we refer to the official codebase [6] and initialize our model from the pre-trained CogVideoX-2B (Yang et al., 2024b) checkpoint. For the schedule of bridge, we adopt the Bridge-gmax schedule of (Chen et al., 2023c), where $f(t) = 0$, $g(t)^2 = \beta_0 + t(\beta_1 - \beta_0)$, $\alpha_t = 1$, $\sigma_t^2 = \frac{1}{2}(\beta_1 - \beta_0)t^2 + \beta_0 t$ with $\beta_0 = 0.01$, $\beta_1 = 50$. We fine-tune the models $\epsilon^{\hat{\Psi}}$ for 20k iterations or 100k iterations with batch size 64. We use the AdamW optimizer with learning rate $1 \times 10^{-5}$ and mixed precision of BFloat16. We do not apply ema to the model weight during fine-tuning. The conditions $c$ and $z^i$ are incorporated into the network in the same way as DynamiCrafter, and we concatenate $\mathbf{z}_t$ with $\mathbf{z}^i$ along the channel or temporal axis to condition the network on the prior (we find that the performance is quite similar whether we conduct the concatenation along channel or temporal axis). As the schedule $\{\alpha_t, \sigma_t\}_{t \in [0,T]}$ is different from that of the pre-trained diffusion models, we use the generalized SAF (Proposition 2).

**Training From Scratch on UCF-101** We reference the codebase of Latte[7] to train FrameBridge from scratch on UCF-101. We adopt Latte-S/2 as our bridge model with the same schedule as above and train FrameBridge for 400k iterations with batch size 40. For FrameBridge with neural prior, we also implement $F_\eta(z^i, c)$ with Latte-S/2 except that the conditioning of timestep $t$ is removed from the model. To match $z^i$ with the input shape of Latte, we replicate $z^i$ for $L$ times and concatenate them along temporal axis. We train $F_\eta(z^i, c)$ for 400k iterations with batch size 32 before training bridge models if the

---
[5]https://github.com/Doubiiu/DynamiCrafter
[6]https://github.com/THUDM/CogVideo
[7]https://github.com/Vchitect/Latte

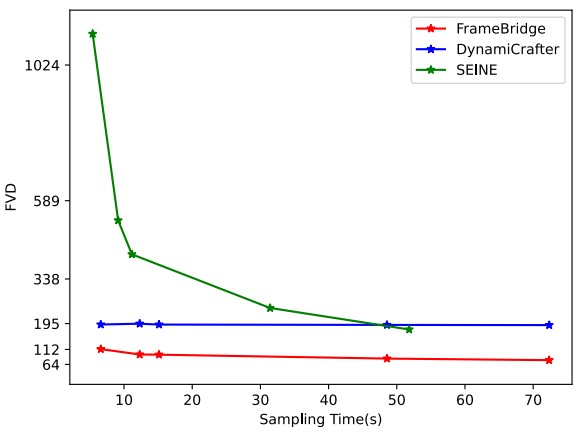

(a) Zero-shot FVD with different execution time

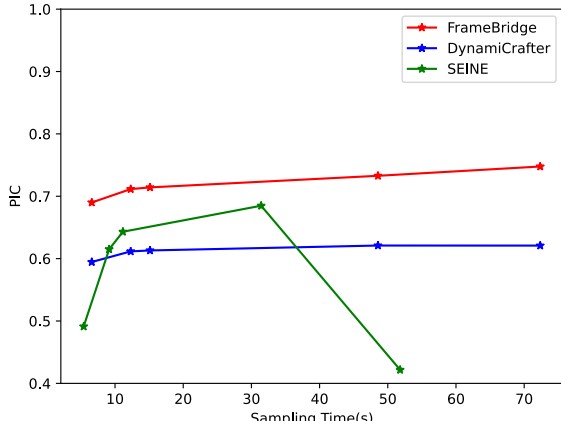

(b) Zero-shot PIC with different execution time

*Figure 8.* Video quality sampled with different execution time.

neural prior is applied. For both the training of bridge models and $F_\eta(z^i, c)$, we use the AdamW optimizer with learning rate $1 \times 10^{-5}$ and ema is not applied. The conditions $c$ are incorporated into the network in the same way as Latte. Since Latte-S/2 is a transformer-based diffusion network, we incorporate the condition $z^i$ by concatenate it with video latent $\mathbf{z}_t$ in the token sequence. To condition the network on prior $\mathbf{z}^i$ or $F_\eta(z^i, c)$, we concatenate them with $\mathbf{z}_t$ along the channel axis.

**SNR-Aligned Fine-tuning**    When implementing the SAF technique, we need to calculate the inverse function of SNR for the teacher diffusion schedule $\tilde{t} = SNR^{-1}(\frac{a_t^2}{c_t^2})$. However, some T2V diffusion models use discrete timesteps and we need to approximate the aligned $\tilde{t}$. In our experiments, we choose to find a discrete timestep $t_n$ such that $SNR(t_n) > \frac{a_t^2}{c_t^2} > SNR(t_{n+1})$ and assume $SNR(\cdot)$ is a linear function with respect to the input $t$ of diffusion schedule in the interval $(t_n, t_{n+1})$ to obtain the aligned $\tilde{t}$.

### D.3.2 BASELINES FOR TEXT-CONDITIONAL I2V GENERATION

For SVD (Blattmann et al., 2023), SEINE (Chen et al., 2023b), ConsistI2V (Ren et al.) and SparseCtrl (Guo et al., 2025), we use the official model checkpoints and sampling code to sample videos for evaluation. For DynamiCrafter (Xing et al., 2024), we sample videos with the official model checkpoints. We also use the official training code [8] to train a DynamiCrafter for 20k iterations with batch size of 64 as a diffusion-based I2V fine-tuning baseline to compare it with FrameBridge fine-tuned under the same training budget.

### D.3.3 BASELINES FOR CLASS-CONDITIONAL I2V GENERATION

**ExtDM (Zhang et al., 2024b)** is a diffusion-based video prediction model, which is trained to predict the following $m$ frames with the given first $n$ frames of a video clip. We train ExtDM with their official implementation[9] and set $n = 1, m = 15$ for our I2V setting on UCF-101.

**VDT-I2V** is our implementation of the I2V method proposed by Lu et al.. They use a transformer-based diffusion network for I2V generation by directly concatenating the image condition with the token sequence of the noisy video latent $\mathbf{z}_t$. We also implement their I2V method on a Latte-S/2 model considering the similarities among transformer-based diffusion models.

---

[8]https://github.com/Doubiiu/DynamiCrafter
[9]https://github.com/nku-zhichengzhang/ExtDM

---

**Algorithm 4** Training algorithms for FrameBridge.

---

**Output:** Trained FrameBridge model $\boldsymbol{\epsilon}_\theta^{\hat{\Psi}}(\mathbf{z}_t, t, \mathbf{z}_T, z^i, c)$.
Set bridge process $\{\alpha_t, \sigma_t, a_t, b_t, c_t\}_{t=0}^T$.
**if** Neural prior is used **then**
  Train a neural prior model $F_\eta(z^i, c)$ with Equation (8) before training FrameBridge.
**end if**
**if** Fine-tuned from pre-trained diffuion model $\boldsymbol{\epsilon}_\phi(\mathbf{z}_t, t, c)$ **then**
  **if** SAF is used **then**
    Re-parameterize the input of $\boldsymbol{\epsilon}_\theta^{\hat{\Psi}}(\mathbf{z}_t, t, \mathbf{z}_T, z^i, c)$ by $\boldsymbol{\epsilon}_\theta^{\hat{\Psi}}(\mathbf{z}_t, t, \mathbf{z}_T, z^i, c) \triangleq \boldsymbol{\epsilon}_{\theta, align}^{\hat{\Psi}}(\tilde{\mathbf{z}}_t, \tilde{t}, \mathbf{z}_T, z^i, c)$ with Equation (24).
    Initialize $\boldsymbol{\epsilon}_{\theta, align}^{\hat{\Psi}}$ with the weight of $\boldsymbol{\epsilon}_\phi(\mathbf{z}_t, t, c)$.
  **else**
    Initialize $\boldsymbol{\epsilon}_\theta^{\hat{\Psi}}$ with the weight of $\boldsymbol{\epsilon}_\phi(\mathbf{z}_t, t, c)$.
  **end if**
**else**
  Randomly initialize $\boldsymbol{\epsilon}_\theta^{\hat{\Psi}}(\mathbf{z}_t, t, \mathbf{z}_T, z^i, c)$.
**end if**
**repeat**
  Sample data $(\mathbf{z}_0, c) \sim p_{data}(\mathbf{z}_0, c)$, timestep $t$ and $\mathbf{z}_t \sim p_{bridge,t}(\mathbf{z}_t|\mathbf{z}_0, \mathbf{z}_T)$.
  Take the first frame of $\mathbf{z}_0$ as the image condition $z^i$.
  **if** Neural prior is used **then**
    $\mathbf{z}_T \leftarrow F_\eta(z^i, c)$.
  **else**
    Construct $\mathbf{z}_T$ by replicating $z^i$.
  **end if**
  $l(\theta) = \left\| \boldsymbol{\epsilon}_\theta^{\hat{\Psi}}(\mathbf{z}_t, t, \mathbf{z}_T, z^i, c) - \frac{\mathbf{z}_t - \alpha_t \mathbf{z}_0}{\sigma_t} \right\|^2$.
  Update $\theta$ with the optimizer and loss function $l(\theta)$.
**until** Reach the training budget

---

### D.3.4 ABLATION STUDIES ON NEURAL PRIOR

In Section 5.3, we ablate on the neural prior technique by comparing the performance of four models:

- **VDT-I2V**: The same model as our diffusion baseline on UCF-101.

- **VDT-I2V with neural prior as the network condition**: The same model as VDT-I2V except that we additionally condition the network on $F_\eta(z^i, c)$.

- **FrameBridge without neural prior**: A FrameBridge model implemented by utilizing the replicated image $\mathbf{z}^i$ as the prior.

- **FrameBridge with neural prior only as the network condition**: A FrameBridge model implemented by utilizing $\mathbf{z}^i$ as the prior. However, we condition the bridge model on $F_\eta(z^i, c)$ by additionally feeding it into the network through concatenation with $\mathbf{z}_t$ along the channel axis.

- **FrameBridge-NP**: A FrameBridge model implemented by utilizing $F_\eta(z^i, c)$ as the prior.

## E  Discussion On Related Works

**Video Diffusion Models**  Inspired by the success of text-to-image (T2I) diffusion models (Ramesh et al., 2022; Nichol et al., 2022), numerous studies have investigated diffusion-based text-to-video (T2V) models (Blattmann et al., 2023; Yang et al., 2024b; Singer et al.) by designing 3D spatial-temporal U-Net (Ho et al., 2022b;a) and Diffusion Transformers (DiT) (Peebles & Xie, 2023; Bao et al., 2023; Zhang et al., 2025e). To improve memory and computation efficiency, Latent Diffusion Models (LDM) (Rombach et al., 2022; Vahdat et al., 2021) are utilized where the diffusion process is applied in

the compressed latent space of video samples (Bao et al., 2024; Brooks et al., 2024; He et al., 2022). Meanwhile, some other works designed cascaded diffusion models to generate motion representation (Yu et al., 2024b) or videos with lower resolution (Ho et al., 2022a; Wang et al., 2025) first, which are utilized to synthesize the result videos in the subsequent stages. Another line of research (Zhang et al., 2025f; Guo et al., 2024; Wu et al., 2023) focuses on leveraging T2I diffusion models to enhance the performance of T2V generation, achieving high spatial quality and motion smoothness at the same time.

**Diffusion-based I2V Generation**  The main difference between I2V and T2V is the incorporation of image conditions into the sampling process. Xing et al. (2024) utilizes the features of a CLIP image encoder and a lightweight transformer to inject image conditions into the backbone of a T2V model. Ma et al. (2024a) and Zhang et al. (2024c) propose to directly model the residual between the subsequent frames and the given initial frame with diffusion for I2V generation. Moreover, Ma et al. (2024a) also uses the DCTInit technique to enhance the consistency of video content with the given image. Chen et al. (2023b) presents to train short-to-long video generation models with masked diffusion models. Guo et al. (2024) and Zhang et al. (2024a) propose to utilize pre-trained T2I models for image animation by training an additional component to model the relationship between video frames. SparseCtrl (Guo et al., 2025) and Animate Anyone (Hu, 2024) design specific fusion modules for video diffusion models to adapt to various types of conditions including RGB images. Ren et al. propose improved network architecture and sampling strategy for image-to-video generation at the same time to enhance the controllability of image conditions. Jain et al. (2024), Zhang et al. (2023) and Shi et al. (2024) design cascaded diffusion systems for I2V generation. VIDIM (Jain et al., 2024) consists of one base diffusion model and another two diffusion models for spatial and temporal super-resolution respectively. Zhang et al. (2023) uses a base diffusion model to generate videos with low resolutions, which serve as the input of the following video super-resolution diffusion model. Shi et al. (2024) first generates the optical flow between the subsequent frames and given image with a diffusion process, and use the optical flow as conditions of another model to generate videos. Ni et al. (2023) and Zhang et al. (2024b) train an autoencoder to represent the motions between frames in a latent space, and use diffusion models to generate motion latents. However, previous I2V diffusion models are built on the *noise-to-data* generation of conditional diffusion process and the sampling remains a denoising process conditioned on given images. In contrast, FrameBridge replaces the diffusion process with a bridge process and the sampling directly model the animation of static images.

**Noise Manipulation for Video Diffusion Models**  Several works have explored to improve the uninformative prior distribution of diffusion models. PYoCo (Ge et al., 2023) recently proposes to use correlated noise for each frame in both training and inference. ConsistI2V (Ren et al.), FreeInit (Wu et al., 2024), and CIL (Zhao et al.)  present training-free strategies to better align the training and inference distribution of diffusion prior, which is popular in diffusion models (Lin et al., 2024; Podell et al.; Blattmann et al., 2023). Noise Calibration (Yang et al., 2024a) proposed to enhance the video quality of SDEdit (Meng et al.)  with iterative calibration of initial noise These strategies focus on improving the noise distribution to enhance the quality of synthesized videos, while they still suffer the restriction of noise-to-data diffusion framework, which may limit their endeavor to utilize the entire information (*e.g.*, both large-scale features and fine-grained details) contained in the given image. In contrast, we propose a *data-to-data* framework and utilize deterministic prior rather than Gaussian noise, allowing us to leverage the clean input image as prior information.

**Comparison with Previous Works of Bridge Models and Coupling Flow Matching**  In Section 4, we leverage the forward SDE of bridge models (Zhou et al.) and the backward sampler proposed by Chen et al. (2023c) to build FrameBridge. We unify their theoretical frameworks to establish our formulation, and emphasize that bridge models are suitable for image-to-video generation, which is a typical *data-to-data* generation task. Liu et al. (2023) and Chen et al. (2023c) apply bridge models to image-to-image translation and text-to-speech synthesis tasks respectively. Similar as bridge models, flow matching can also be used to construct the data-dependent stochastic interpolants (Albergo et al., 2024; Fischer et al., 2023; Albergo et al., 2023) for paired-data generation and has been used in image-to-image generation. However, whether the coupling flow matching is suitable for image-to-video generation has not been fully explored. Compared with their works, we focus on I2V tasks, building our bridge-based framework by utilizing the *frames-to-frames* essence and presenting two innovative techniques for two scenarios of training I2V models, namely fine-tuning from pre-trained text-to-video diffusion models and training from scratch.

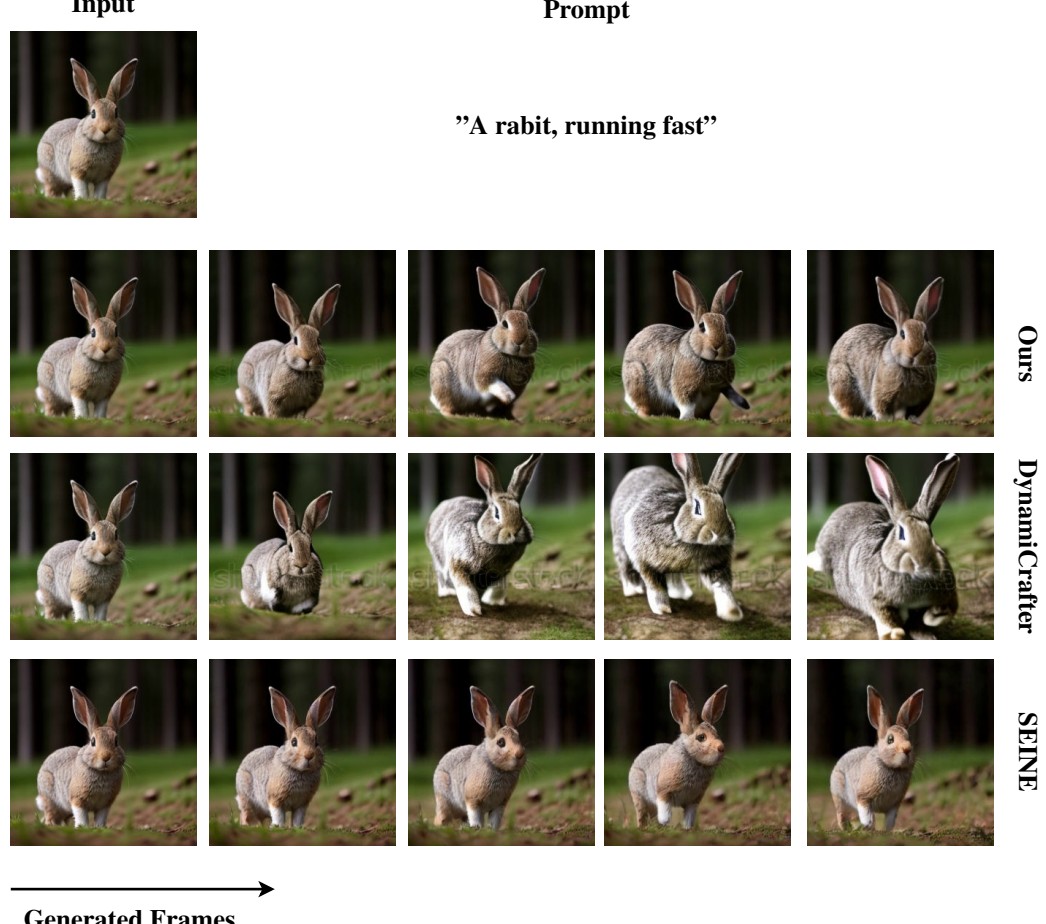

*Figure 9.* **Another case of qualitative comparison between FrameBridge and other baselines.** FrameBridge outperforms diffusion baseline methods in appearance consistency and video quality. FrameBridge and DynamiCrafter models are fine-tuned from VideoCrafter1 for 20k steps.

## F   More Qualitative Results of FrameBridge

We show several randomly selected samples of FrameBridge below, and more synthesized samples can be visited at: https://framebridge-icml.github.io/

**Input**

**Prompt**

a blue fishing boat is navigating in the ocean next to a cruise ship

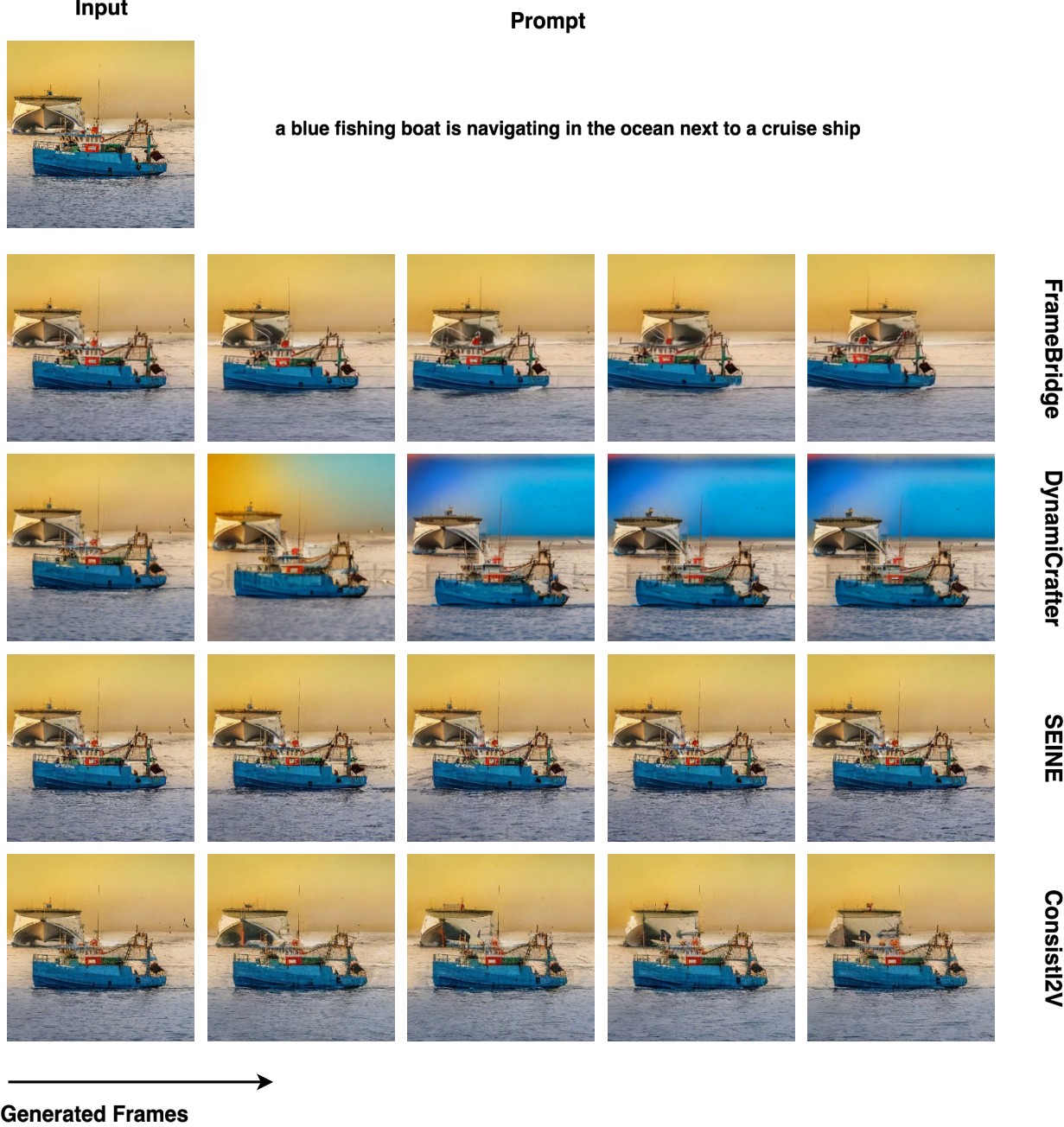

Generated Frames

*Figure 10.* **Qualitative comparison between FrameBridge and other baselines.** Here FrameBridge is fine-tuned from CogVideoX-2B for 100k steps, and the samples of other baselines are generated with their official checkpoints. DynamiCrafter, SEINE, ConsistI2V are fine-tuned from VideoCrafter1, inflated Stable Diffusion 2.1-Base and LaVie respectively.

**Input**

**Prompt**

**a bridge that is in the middle of a river, camera zooms out**

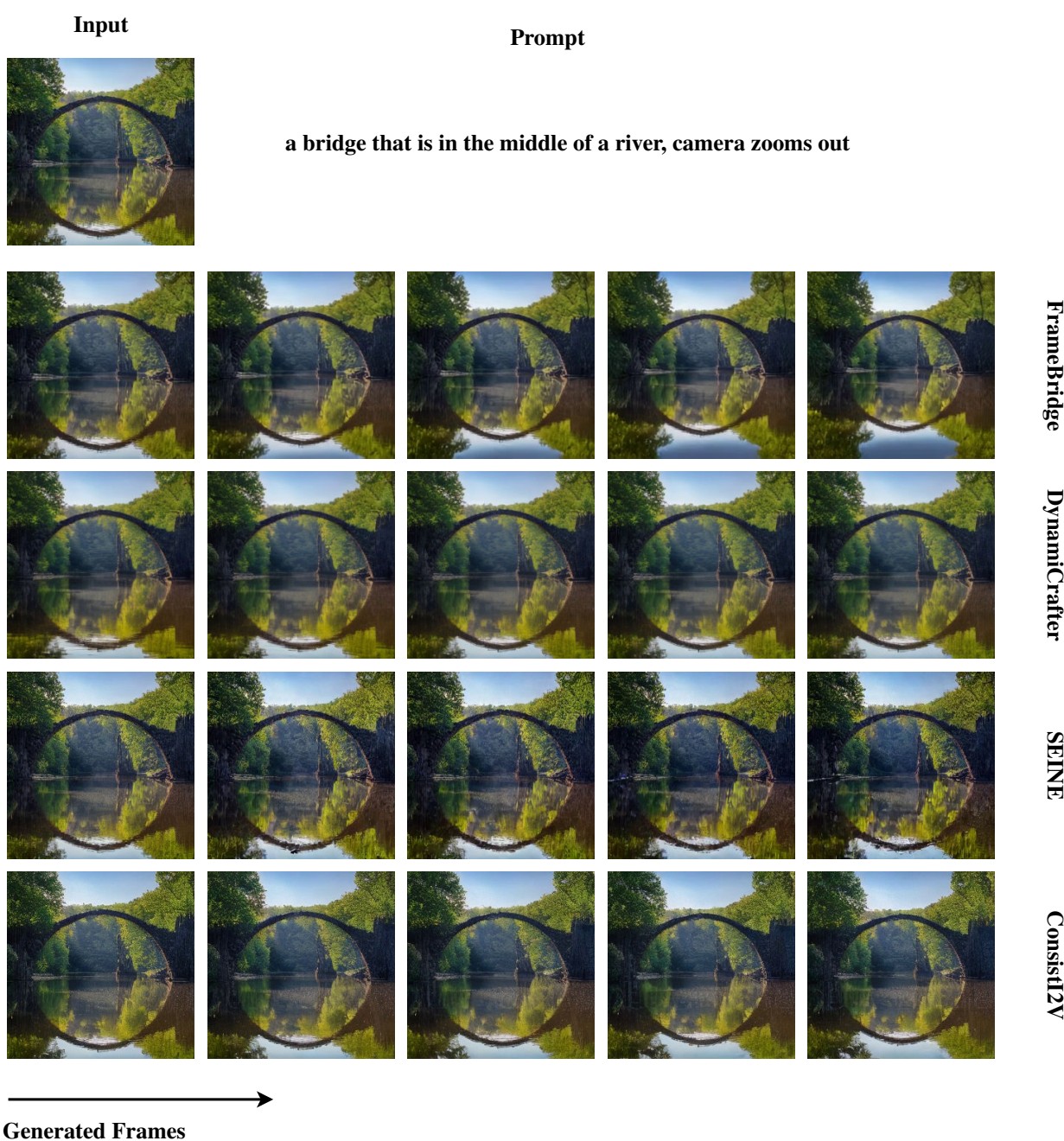

**Generated Frames**

*Figure 11.* **Qualitative comparison between FrameBridge and other baselines.** Here FrameBridge is fine-tuned from CogVideoX-2B for 100k steps, and the samples of other baselines are generated with their official checkpoints. DynamiCrafter, SEINE, ConsistI2V are fine-tuned from VideoCrafter1, inflated Stable Diffusion 2.1-Base and LaVie respectively.

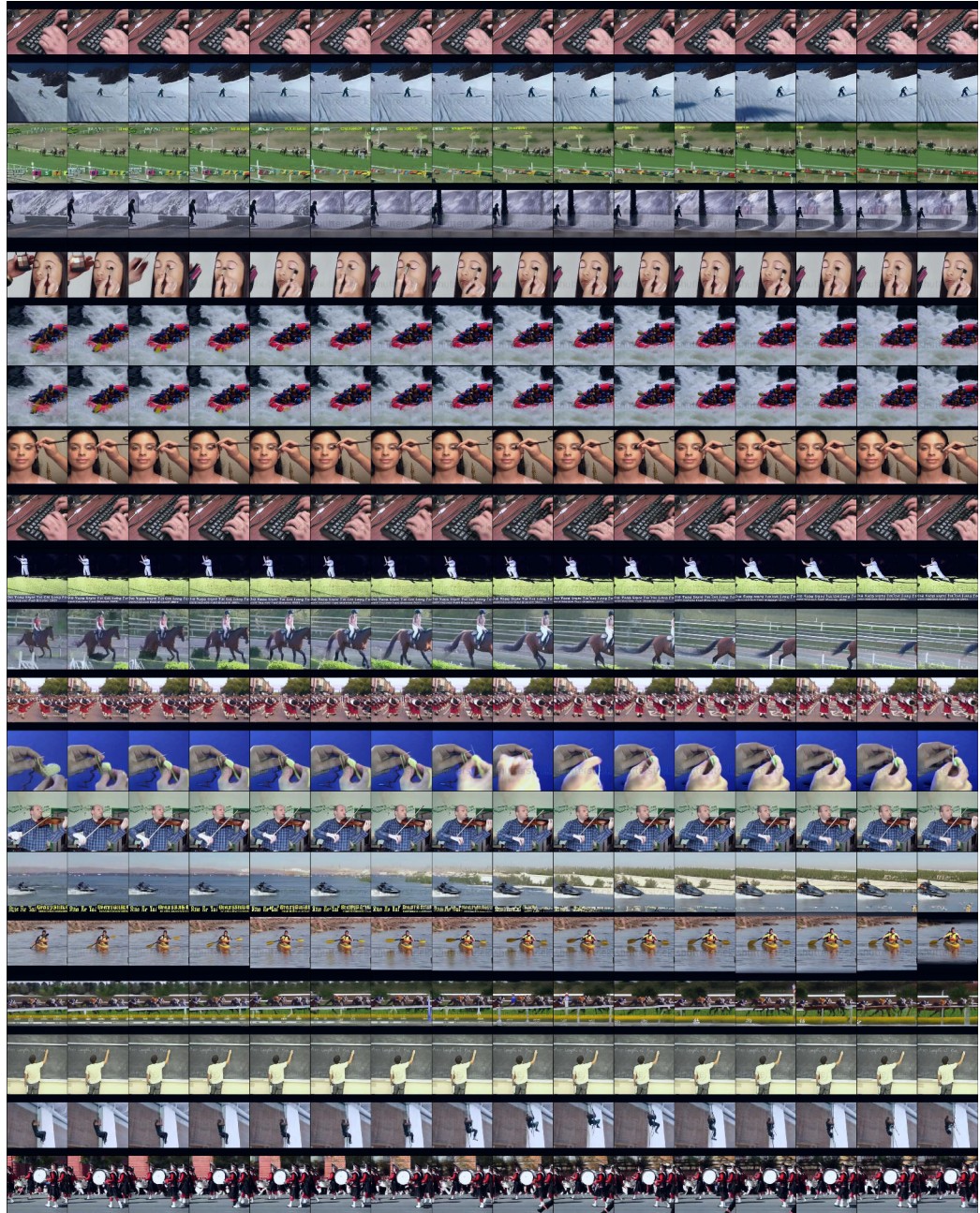

*Figure 12.* Zero-shot generation results of fine-tuned FrameBridge (with SAF) on UCF-101.

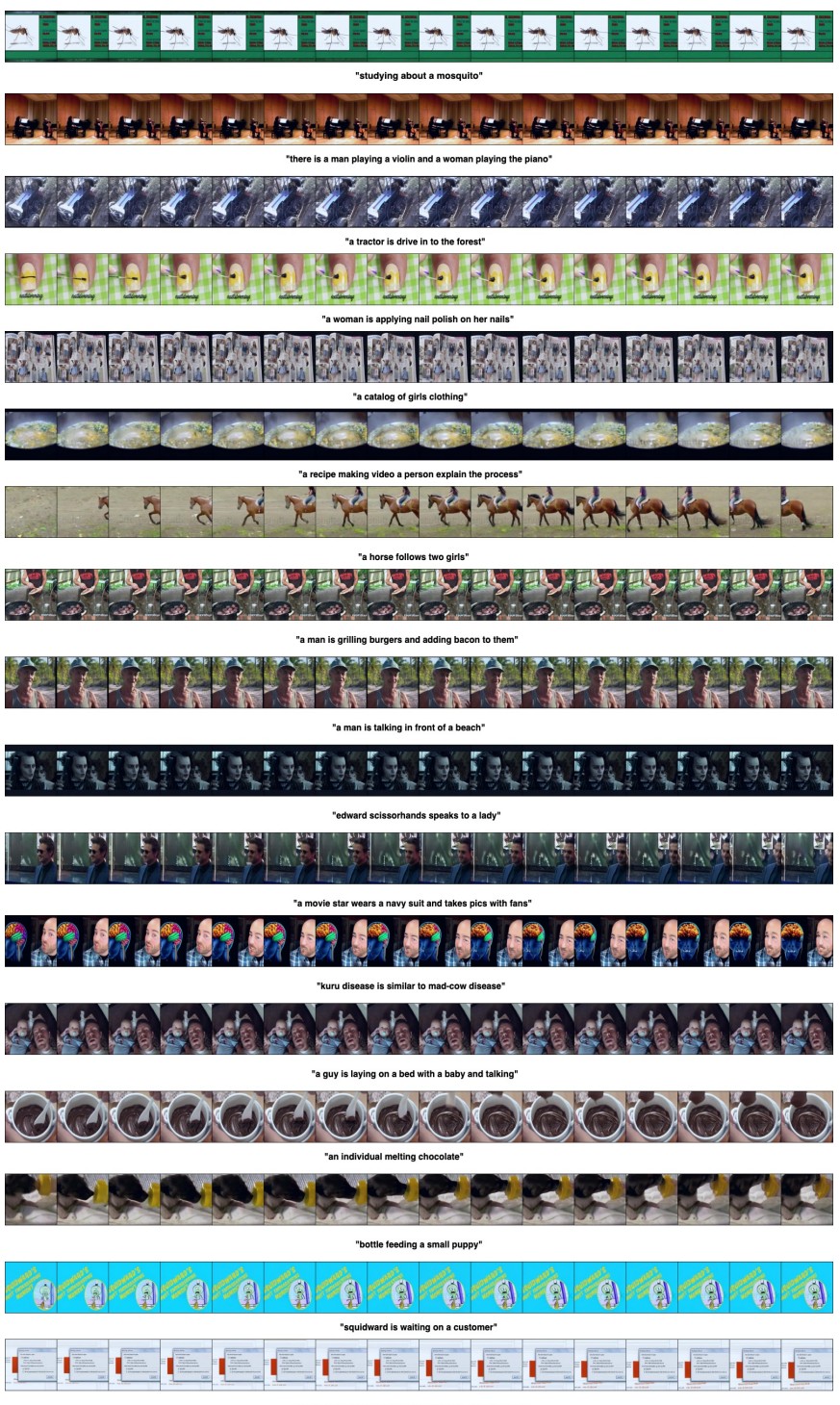

*Figure 13.* Zero-shot generation results of fine-tuned FrameBridge (with SAF) on MSR-VTT.

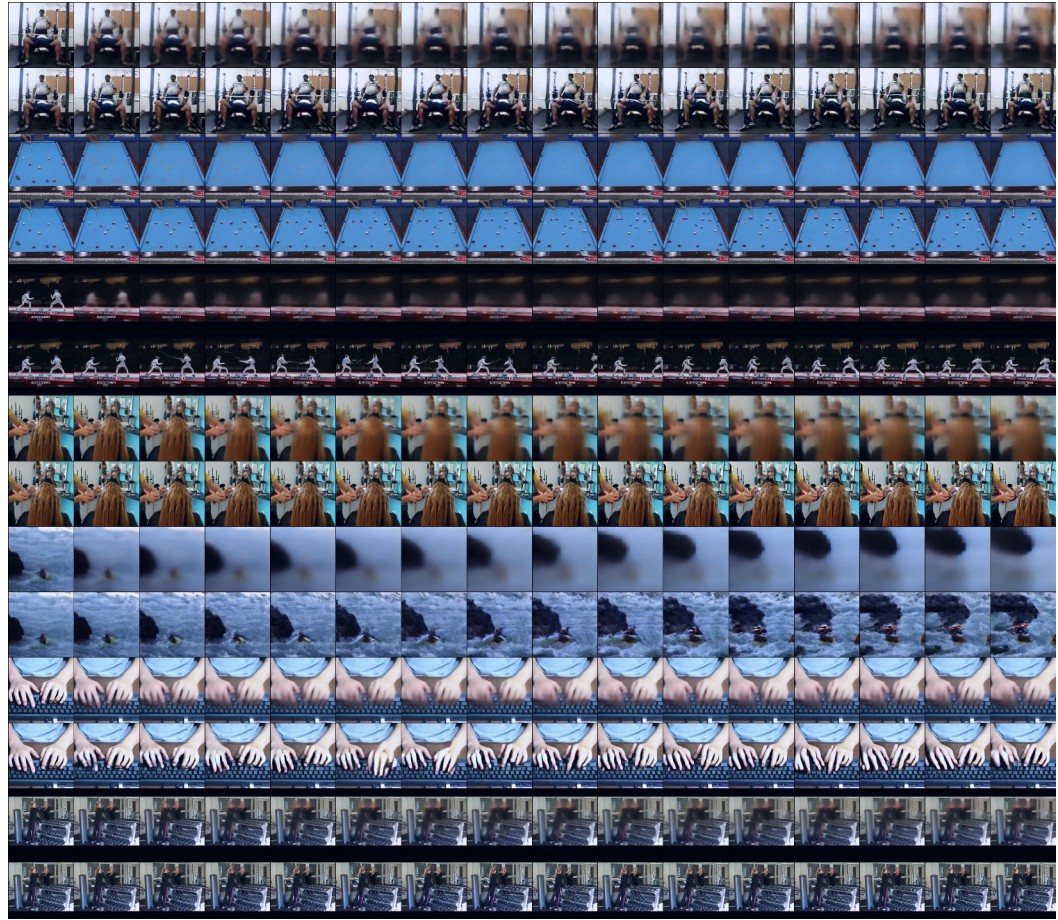

*Figure 14.* Non-zero-shot generation results of FrameBridge-NP on UCF-101. We use two lines to present a neural prior and the corresponding generated video.

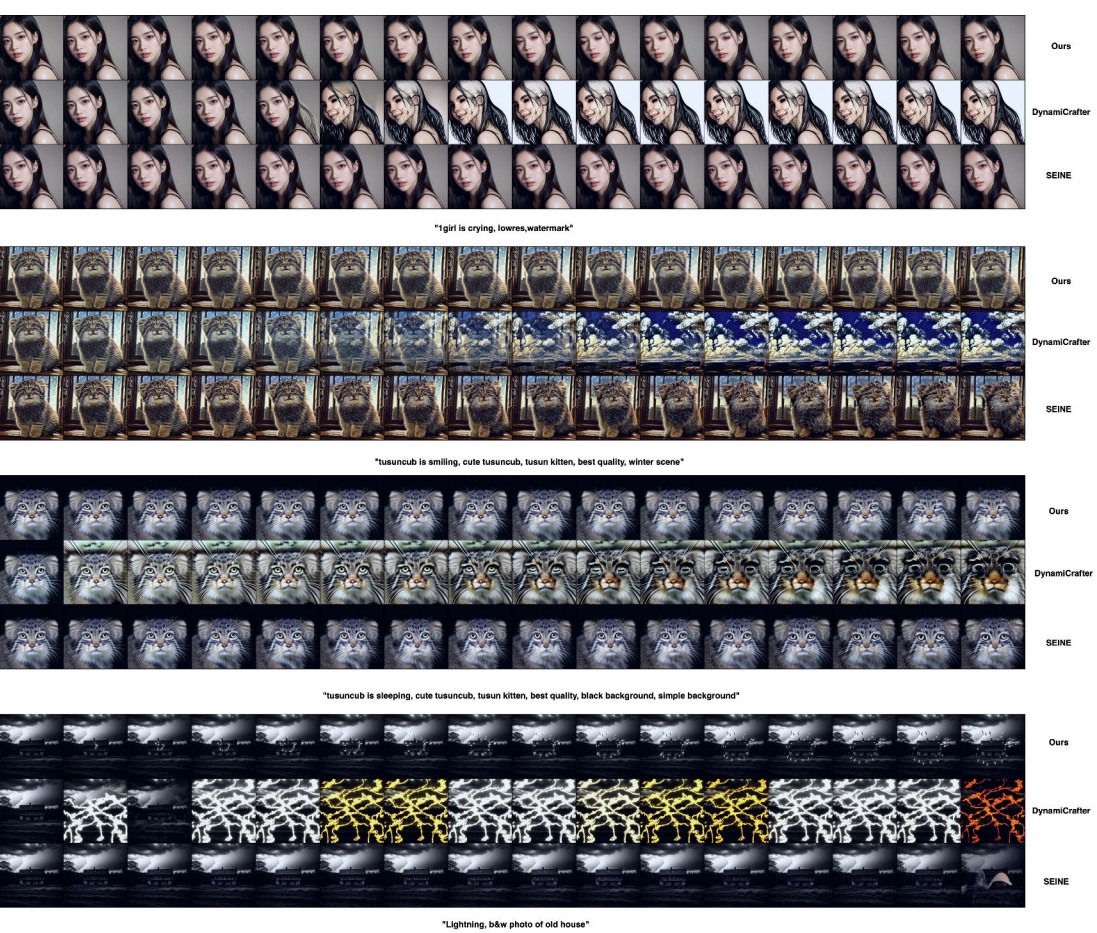

*Figure 15.* Comparisons between fine-tuned FrameBridge and other diffusion-based I2V models.

