# OpenReview forum: "FrameBridge: Improving Image-to-Video Generation with Bridge Models"
_ICML.cc/2025/Conference — ICML 2025 poster_

### Official Review · Reviewer_2c9b · 2025-03-08

**Overall Recommendation:** 3

**Summary:**

This paper introduces FrameBridge, a novel approach to improve image-to-video (I2V) generation using diffusion models. The authors address the mismatch between the noise-to-data generation process of traditional diffusion models and the I2V task, which can lead to suboptimal results. FrameBridge proposes a bridge model that adopts a data-to-data generative process, enabling better utilization of the given image and enhancing consistency in video generation. The paper also introduces two innovative techniques: SNR-Aligned Fine-tuning (SAF), which adapts pre-trained text-to-video models for I2V tasks, and a neural prior to further improve the model when training from scratch. Experimental results on WebVid-2M and UCF-101 demonstrate that FrameBridge outperforms existing diffusion models in synthesis quality, with significant improvements in FVD scores. The SAF and neural prior techniques also contribute to the enhanced performance of bridge-based I2V models.

**Claims And Evidence:**

Yes

**Essential References Not Discussed:**

[1] VideoElevator: Elevating Video Generation Quality with Versatile Text-to-Image Diffusion Models. arXiv:2403.05438
[2] Noise Calibration: Plug-and-play Content-Preserving Video Enhancement using Pre-trained Video Diffusion Models. arXiv 2407.10285

These two papers also investigate how to improve the performance of various video diffusion models, and the authors are encourgaed to discuss them in this paper.

**Experimental Designs Or Analyses:**

Yes.

**Methods And Evaluation Criteria:**

Yes

**Other Comments Or Suggestions:**

1. The authors are encouraged to provide more video comparisons in supplementary materials and specify the name of pre-trained video diffusion models.
2. The number of decimal places in the table should be kept consistent as much as possible, e.g., Table 1 and Table 2.

**Other Strengths And Weaknesses:**

1. The technical contributions are somewhat limited. The proposed FrameBridge seems like a simple combination of video diffusion models and bridge models.
2. The paper only provides qualitative experimental comparisons with earlier video generation models (e.g., DynamiCrafter), but does not include comparisons with the latest models (e.g., CogVideoX, HunyuanVideo).
3. In Fig.1, the key idea of FrameBridge is to take duplicated images as the initial video. The reviewer is curious that: what is the effect of directly adding noise to the initial video and using a video diffusion model to denoise it? Will FrameBridge perform better than this baseline?

**Questions For Authors:**

Please see weaknesses.

**Relation To Broader Scientific Literature:**

This paper mainly focuses on improving the appearance and temporal consistency of video diffusion models and has shown its effectiveness on various models.

**Theoretical Claims:**

Yes. I checked the basics of DDBM, parameterization of FrameBridge, SNA-Aligned Fine-tuning and training objective in Appendix.

---

> ### Author Rebuttal · Authors · 2025-04-01
>
> Dear Reviewer 2c9b,
>
> We sincerely appreciate your recognition of the strengths and effectiveness of our work and the valuable suggestions to help us improve that. We are happy to have a discussion and hope it could address your concerns. Tables are provided in https://framebridge-icml.github.io/rebuttal-demo-page/.
>
> ## Contributions of FrameBridge
>
> **1. Reply to W1**
>
> We would like to gently point out that the contribution of our work is not limited and our FrameBridge is a novel bridge-based I2V framework:
>
> (1) **SAF technique: Applying bridge to I2V generation is not trivial.** As shown by Table 4 of our paper, directly fine-tuning **T2V diffusion** to **I2V bridge** may cause suboptimal video quality, and we propose SAF technique to align the two process. **As far as we know, it is the first trial to fine-tune bridge models from diffusion models.** A concurrent work [13] also shows the importance of aligning different processes in fine-tuning. **Table 3, 4 in our link** provide more evidence for the effectiveness of SAF.
>
> (2) **We propose neural prior to further improve the performance of bridge model.** As bridge-models benefit from informative priors, it is well-motivated and effective to further investigate the choice of prior.
>
> (3) We provide theoretically sound proof and derivations for all the techniques we proposed.
>
> **In conclusion, we are the first to apply bridge models to I2V generation, which may not achieve superior generation quality without the two novel proposed techniques (i.e., SAF and neural prior).**
>
> **2. Discussions about other related works**
>
> We thank the reviewer for pointing out the related works we unintentionally left out, and will add citations and discussions in our revised paper. VideoElevator [14] proposed to improve the T2V generation quality with T2I diffusions, achieves high spatial quality and motion smoothness at the same time. Noise Calibration [15] proposed to enhance the video quality of SDEdit [16] with iterative calibration of initial noise. These are valuable works investigating into the improvement of T2V diffusion models, laying foundations for better I2V generation by building enhanced base models and providing viable techniques for improving diffusion-based I2V generation.
>
> ## Qualitative Comparisons
>
> **Reply to S1 and W2:** We thank the reviewer for the instructive and valuable suggestions and add a later academic baseline ConsistI2V (TMLR 2024) [7] in our qualitative comparisons. Although we fine-tune a FrameBridge model from CogVideoX, we choose CogVideoX-2B as base model and fine-tune for several hundred GPU hours, which is $< 1\%$ of that used to train CogVideoX-I2V (fine-tuned from CogVideoX-5B), HunyuanVideo and other industry-grade models. We provide more qualitative comparisons in https://framebridge-icml.github.io/rebuttal-qualitative/, and pre-trained models are specified. Please feel free to tell us if you still have any concern, and we are willing to provide more experimental evidence or have a further discussion.
>
> ## Comparing FrameBridge with Inference-Time Noise Manipulation
>
> **Reply to W3:** If we understand correctly, the provided baseline is a inference-time noise manipulation which only changes the prior when sampling and do not modify the diffusion process. (We are not certain if we have understood correctly. Feel free to point it out if there is misunderstanding and we are willing to provide more experimental resutls.) This baseline will generate almost static videos, and thus the quality is much inferior to FrameBridge and diffusion I2V models.
>
> Table 1. Zero-shot metrics on MSR-VTT and VBench-I2V score.
> |Model|FVD|CLIPSIM|PIC|VBench-I2V Score|
> |-|-|-|-|-|
> |Baseline|644|0.2249|0.56|77.36|
> |FrameBridge|**99**|**0.2250**|**0.70**|**85.37**|
>
> Due to character limit, we kindly invite you to read our response to Revewer fXNN (**Comparison between bridge model and inference-time noise manipulation** part), where we provide more intuitive explanations. In case that you are interested in the comparison between bridge models and flow-matching (which shares some similarities with the mentioned the baseline), we kindly invite you to read our response to Reviewer g6Xo (**Comparing bridge and flow-matching** part).
>
> ## Other Discussions
> **Reply to S2**: Thanks for the suggestion and we will revise that in out paper later. (In this response, we temporarily align the decimal places with the current version.)
>
> Due to the character limit, we can not elaborate on all the details. Please feel free to tell us if you still have any concern or question. We are willing to continue to have a kind and thorough discussion.
>
> References: See response to Reviewer g6Xo.

---

### Official Review · Reviewer_g6Xo · 2025-03-12

**Overall Recommendation:** 4

**Summary:**

This paper introduces FrameBridge, reformulating the image-to-video task as data-to-data generation through a bridge model. Different from image-to-video as first frame condition (i.e., noise-to-data generation), data-to-data generation achieves better consistency and can well preserve information in the first frame. Besides, the authors further introduce SNR-Aligned fine-tuning and neural prior to improve the performance of FrameBridge.

**Claims And Evidence:**

The experiments in Table 4 and Table 5 demonstrate part of the claim. However, the authors do not directly compare noise-to-data and data-to-data in the ablation studies. Although the authors compare their approach to other noise-to-data paradigms, like ExtDM, the training settings and architectures are not matched.

**Essential References Not Discussed:**

N/A.

**Experimental Designs Or Analyses:**

I think most of the experimental designs are soundness. However, it is better to directly compare noise-to-data and data-to-data in ablation study.

**Methods And Evaluation Criteria:**

Yes.

**Other Comments Or Suggestions:**

1) The author should discuss/experiment with flow-matching, since flow-matching can also be adopted to data-to-data transfer. Is it any benefit to use bridge model than flow-matching?

**Other Strengths And Weaknesses:**

Strengths: 1) FrameBridge is well-motivated and the neural prior is a novel idea for initializing data distribution.
Weaknesses: 1) The authors should carefully ablate noise-to-data and data-to-data paradigms with fair training settings. Does data-to-data learn a shortcut to more static motion? It would be better to analyze the differences between these two paradigms to provide more insights for readers.

**Questions For Authors:**

N/A.

**Relation To Broader Scientific Literature:**

The contributions are broadly related to distribution transfer between arbitrary two data distributions. This paper resorts to a bridge model to translate data-to-data distribution, while this idea can also be implemented with flow-matching.

**Theoretical Claims:**

Yes.

---

> ### Author Rebuttal · Authors · 2025-04-01
>
> Dear Reviewer g6Xo,
>
> We sincerely appreciate your acknowledgement of the strengths of our work and instructive suggestions to help us improve that. We hope the following discussions could address your concerns and questions. Tables are provided in https://framebridge-icml.github.io/rebuttal-demo-page/.
>
> ## Ablations between noise-to-data and data-to-data
>
> We thanks the reviewer's suggestion and provide the experimental results of this ablation.
>
> 1. For the setting of fine-tuning, we provide the results in **Table 3, 4**. Here we use the same base model, network architecture, training data, training budget for noise-to-data and data-to-data models. As we need to fine-tune bridge models from pre-trained diffusion models, it is important to align the two process with our proposed SAF technique. A concurrent work [13] reaches similar conclusions when fine-tuning from flow-matching to diffusion process.
>
> 2. For the setting of training-from-scratch, since VDT-I2V has the same network architecture (Latte-S/2), training dataset, training budget as FrameBridge, we would like to gently point out that Table 3, 5 in the paper can offer experimental evidence for the ablation between noise-to-data and data-to-data. We appreciate the reviewer's question which help us to identify the lack of clarity in the setup (details are now dispersed in Sec. 5.2, App. D.3.3, D.3.4) and will revise it for better clarity.
>
> ## Does data-to-data learn a shotcut to more static motion?
>
> This is a thoughtful question. Bridge models will not learn a shortcut to more static motion. Due to character limit, we kindly invite you to read our response to Reviewer fXNN for more discussions (**Reply to Q2** part).
>
> ## Comparing noise-to-data and data-to-data
>
> We wold like to offer more intuitive comparisons between the two frameworks.
>
> 1. When considering the differences between noise-to-data and data-to-data frameworks, we suggest to view from the perspective of forward-backward process instead of pointwise denosing. Theoretically, denoising score matching guarantees perfect score function for each $(z_t, t)$. But practically the score function is not perfect for each point, and we should consider the complexity for modeling the whole process.
>
> 2. Differences in process: For noise-to-data I2V generation, the image can only be injected into the model by network conditioning. For data-to-data I2V generation, the whole diffusion process is changed to a bridge process and now we can utilize the image condition from two aspects (network conditioning and process). As shown by Figure 2 in our paper, the marginal distribution $z_t$ in bridge process carries more information than in diffusion. In alignment with our experiments, **modeling the whole bridge process** with bridge score function could be easier than **modeling the whole diffusion process** with diffusion score function.
>
> We hope the discussion could provide some intuitive insights for the readers and are willing to have a further discussion if there are other cocerns or questions.
>
> ## Comparing bridge and flow-matching
>
> We conduct experiments with 3 flow-matching models and compare them with diffusion and FrameBridge:
>
> 1. Vanilla flow-matching: The noisy latent takes the form of $z_t = (1-t)z_0 + t \epsilon, \epsilon \sim \mathcal{N}(0, I)$
> 2. Coupling flow-matching with $\sigma=1.0$ [18]: $z_t = (1-t)z_0 + t (z^i + \epsilon)$
> 3. Coupling flow-matching with $\sigma=0$ [19]: $z_t = (1-t)z_0 + t z^i$
>
> Results are shown in **Table 12** of our link. We compare bridge and flow-matching in both methods and experiment results as below:
>
> 1. The algorithm and performance of Vanilla flow-matching is quite similar to that of diffusion.
> 2. Coupling flow-matching with $\sigma=0$ will generate almost static video. In [19], it is used to refine the results of a diffusion, and need condition augmentation, which is not suitable for single-stage I2V.
> 3. Coupling flow-matching with $\sigma=1$ has better performance than diffusion models. Different from bridge models, the prior is still a Gaussian (with non-zero mean), while the prior of bridge is a deterministic point. Empirically, as shown by the experiment results, brige models can utilize the condition more effectively.
>
> Due to character limit, we can not elaborate on all the details. Please feel free to tell us if you still have any concern and we are willing to have a further discussion.
>
> References (Due to character limit, we can only provide abbreviated forms of some works where there is no ambiguity):
>
> [1] VBench++
>
> [2] Stable Video Diffusion
>
> [3] CogVideoX
>
> [4] Conditional Image Leakage
>
> [5] I^2SB
>
> [6] DDBM
>
> [7] ConsistI2V
>
> [8] RPGDiffusion
>
> [9] MovieDreamer
>
> [10] VideoGen-of-Thought
>
> [11] VideoDirectorGPT
>
> [12] VIDIM
>
> [13] SANA-Sprint
>
> [14] VideoElevator
>
> [15] Noise Calibration
>
> [16] SDEdit
>
> [17] Flow Matching for Generative Modeling
>
> [18] Stochastic Interpolants with Data-Dependent Couplings
>
> [19] Boosting Latent Diffusion with Flow Matching

---

### Official Review · Reviewer_WAVy · 2025-03-12

**Overall Recommendation:** 3

**Summary:**

This paper proposes a bridge model-based image-to-video generation model. It first formulates image-to-video generation as data-to-data generation instead of noise-to-data generation. Under this formulation, the generation should be easier because it starts from a strong prior of the image instead of the Gaussian prior. Furthermore, two designs are proposed to help finetune a text-to-video model for image-to-video generation: an SNR-aligned parameterization and a neural prior that predicts the mean image using a neural network. The proposed method is evaluated on WebVid-2M, UCF-101 and VBench-I2V, showing advantageous performance over different base models.

## update after rebuttal
The authors have addressed all of my concerns with clarifications and experimental results. Therefore, I am raising my score to Weak accept.

**Claims And Evidence:**

The paper makes three main claims: 1) image-to-video generation can be formulated as a data-to-data generation task, 2) SNR-aligned parameterization helps finetune a text-to-video model to image-to-video generation, 3) neural prior helps train the image-to-video model from scratch. While the first two claims are supported by extensive experimental evidence, I have concerns about the third claim:

* **W1**: The predicted neural prior looks blurry (as shown in Figure 4), which could lead to inferior detail in the generated video. The authors seem to adopt an implementation of concatenating the neural prior with the original image to preserve the details (the second row of Figure 4), but this is more of a hack and weakens the contribution of this design.

**Essential References Not Discussed:**

There are a few seminal works on bridge models that are not cited [De Bortoli'21, Peluchetti'21].

---

[1] De Bortoli, et al. Diffusion Schrödinger Bridge with Applications to Score-based Generative Modeling. NeurIPS 2021.

[2] Peluchetti. Non-Denoising Forward-Time Diffusions. 2021.

**Experimental Designs Or Analyses:**

The experimental designs could be improved in the following directions:

* **W2**: Inconsistent base model and setup across experiments. The current ablation studies use a different base model (VDT-I2V) and setup (non-zero-shot UCF-101) than the main experiments, which is confusing. It would be better if the authors could perform ablation studies using the same base model and setup (i.e., DynamicCrafter with zero-shot UCF-101). Even more, it is unclear to me whether the DynamicCrafter baseline is properly initialized in the main experiments, since the convergence curves in Figure 6 do not seem to share the same starting point.
* **W3**: Evaluate on long video generation. The current method is evaluated only on 16-frame (2s) video generation, where the proposed strong image prior is clearly beneficial, as short videos don't deviate much from the starting frame. However, long video generation (e.g. $\ge$ 5s) requires the generation of new content that is significantly different from the starting frame, where a strong image prior could become a limitation. It would be better if the authors could investigate its effectiveness across different generation lengths.

**Methods And Evaluation Criteria:**

The proposed method is suitable for image-to-video generation because it translates the noise-to-data generation problem into an easier data-to-data generation form. The evaluation criteria are also quite comprehensive, including FVD, IS, and PIC on UCF-101, FVD, CLIPSIM, and PIC on MSR-VTT, and various metrics in VBench-I2V.

**Other Comments Or Suggestions:**

Judging from the demo samples provided on the website, the generated videos do not have a high dynamic degree or large camera motion. It would be better if the authors could include more demo samples with high motion.

**Other Strengths And Weaknesses:**

Strengths:

* The idea of applying the bridge model to image-to-video generation is novel and elegant. It reduces the gap between prior distribution and target distribution in video generation.
* The theoretical analysis on SNR-aligned parameterization and neural prior enhances the completeness of the paper.

Weaknesses:

* **W4**: The proposed method shows a much inferior dynamic degree on VBench-I2V (48.29 vs. 81.95). This is a fundamental limitation associated with the proposed strong static image prior. The authors should carefully discuss this limitation and propose possible solutions.
* **W5**: The results on VBench-I2V seem to be different from the original paper [Huang'24] or the official website. I suggest the authors further check the evaluation protocols and make sure that the baseline results are consistent.

---

[1] Huang, et al. VBench++: Comprehensive and Versatile Benchmark Suite for Video Generative Models. 2024.

**Questions For Authors:**

* **Q1**: Intuitively, the difference between noise-to-data generation and data-to-data generation is not that great, since the model outputs differ only by a constant offset value $z^i$, which is given in the image condition. It should not be difficult for the model to copy the image value $z^i$ and add it to its output. What do you think contributes to the performance gap, is it due to the closer value range of source and target latents, which makes it more friendly for the neural network?
* **Q2**: How does the model handle scene cuts, since it imposes such a strong static image prior?

**Relation To Broader Scientific Literature:**

The paper leverages the diffusion bridge model [Zhou’23] for image-to-video generation, extending its previous success in image-to-image translation [Liu’23, Zhou’23].

**Theoretical Claims:**

The paper makes two theoretical claims: 1) SNR-aligned parameterization aligns with a VP diffusion process, 2) the neural prior actually predicts the mean value of subsequent frames. The proofs of these two claims are provided in Appendix A, and the proofs look solid.

---

> ### Author Rebuttal · Authors · 2025-04-01
>
> Dear Reviewer WAVy,
>
> We sincerely appreciate your recognition of the strengths of our work and providing valuable suggestions to help us improve it. We hope the following discussions can address your concerns. Tables and demos are provided in https://framebridge-icml.github.io/rebuttal-demo-page/.
>
> ## Performance on VBench-I2V
>
> **1. Reply to W5**
>
> We find there are discrepancies between the results of Table 2 in our paper and the latest version of VBench-I2V [1] and thank the reviewer for bringing this to our attention. We present the modified version in **Table 1**. Note that **two FrameBridge models still achieve higher total score than baselines**.
>
> **2. Reply to W4**
>
> It is a valuable consideration and we will provide our empirical results and theoretical discussions. Before that, we would like to gently point out that the much higher DD of SparseCtrl may lead to inferior overall video quality as shown by **Table 1 and 6**. We kindly invite you to read the discussions in our response to Reviewer fXNN (**Reply to Q2** part) due to the character limit.
>
> We would appreciate it if you could kindly take the time to read that, and it may also provide intuitions about why the samples can "deviate" from the prior, which is also appicable to the scenarios involved in W1 and W3.
>
> ## Implementation Details
> **Reply to W1**
>
> Firstly, we would like to calrify that neural prior predicts the **mean video conditioning on the first frme** instead of "mean image". (i.e., Frames can be different.) We list the model input, condition, priors of diffusion and bridge models below.
> |Model|Input|Condition|Prior|
> |-|-|-|-|
> |Diffusion|$z_t,t,c,z^i$|$c, z^i$|$z_T\sim\mathcal{N}(0, I)$|
> |Bridge|$z_t,t,c,F_\eta(z^i, c)$|$c, z^i,F_\eta(z^i, c)$|$z_T=F_\eta(z^i,c)$|
>
> Here $F_\eta(z^i, c)\in \mathbb{R}^{F \times h \times w \times d}$ has the same shape as video latents and we do not concatenate $F_\eta(z^i, c)$ with the original image when it serves as a prior (as illustrated in our pseudo code, line 846), and this design is not a hack. However, both diffusion and bridge model should take original image as an input to ensure it is learning the conditional score function. Blurry prior will not degrade quality of final videos (as shown by our experiments) and bridge models here can be seen as a refiner. (e.g., Bridge can be applied to deblurring [5].)
>
> ## Experimental Design
>
> **1. Reply to W2**
>
> To address the concern, we conduct ablations of SAF in zero-shot setting and provide results in **Table 3, 4**. Neural prior is designed for training from scratch and the ablation setting is aligned with the main experiments in our paper.
> We would like to gently point out that the ablation with VideoCrafter is already provided in the Table 1 of paper (line 342, 344, 345). We thank the reviewer's suggestion and will improve the clarity of ablation studies. For Figure 6, the first point is evaluated when the model is fine-tuned for 3k steps. When initialized, both models generate noise and thus we omitted it. To address the concern, we provide the CD-FVD curve before 3k in **Figure 1**.
>
> **2. Reply to W3**
>
> This is a thoughtful suggestion to help us show the effectiveness of FrameBridge more thoroughly. Due to the constraints of time and computational resources, we temporarily provide the experiment results with 24 frames in **Table 5**.
> Conducting experiments with long video requires significant computational resources: The base model should be large enough to have capability to deal with long video and training with more frames will also add to computational complexity. As far as we know, baselines and benchmarks under this setting (I2V for longer than 16 frames) is not mature enough. Taking all the above into account, we compare FrameBridge with a diffusion I2V model fine-tuned from the same CogVideoX-2B, and find FrameBridge outperforms the diffusion counterpart. Please feel free to tell us if you still have concerns and we are willing to provide more experimental evidence and discussions.
>
> ## Other Discussions
> **1. Reply to Q1**: This is an insightful question and we think the input/output parameterization is not the most important difference. We kindly invite you to read our response to Reviewer g6Xo (**Comparing noise-to-data and data-to-data** part.)
>
> **2. Reply to Q2**: We kindly invite you to read our response to Reviewer fXNN (**Reply to Q1** part).
>
> **3. Additional references**: We thank the reviewer for pointing out the related works we unintentionally left out, and will add citations and discussions in our paper later.
>
> **4. More demos of FrameBridge**: We provide more demos generated by FrameBridge in **Table 9-11**, which we hope could address the concern of dynamic degree.
>
> Due to the word limit, we could only elaborate on some important points. Please feel free to tell us if you still have any concerns and we would be happy to have a kind and thorough discussion further.
>
> References: See response to Reviewer g6Xo.

---

> > ### Comment · Reviewer_WAVy · 2025-04-01
> >
> > Thank you for the detailed response. However, I still have a few remaining concerns:
> >
> > 1. Related to W4: Since the DD metric is not very reliable as you suggested, a user study would be necessary to confirm that there is no disadvantage in terms of dynamic degree. Additionally, the comparison in Table 6 on the rebuttal website between FrameBridge-CogVideoX and previous baselines is not entirely fair, as the improvement may come from differences in the base model rather than the method itself.
> > 2. Related to Q1: The advantage of bridge models over **conditional** diffusion models is not so clear to me. Specifically, in your response to Reviewer g6Xo, you mention that "the marginal distribution $z_t$ in bridge process carries more information than in diffusion". However, $z_t$ in the bridge model is an interpolation of the source distribution, target distribution, and noise, which doesn't necessarily contain more information than in conditional diffusion models where the clean condition is directly encoded.
> > 3. Related to Q2: Using LLMs planners to handle scene cuts may not be a good solution, since some transitions are smooth rather than abrupt.
> >
> > **Update**: Thank you for addressing my concerns, I have raised my score accordingly.

---

> > > ### Author Response · Authors · 2025-04-06
> > >
> > > Dear Reviewer WAVy,
> > >
> > > We appreciate your time in reading our response and engaging in the constructive discussions. We would like to provide additional experimental evidence and related discussions, and we hope this could address your remaining concerns.
> > >
> > > ## Related to W4
> > >
> > > To address the concern, we conduct a user study to compare with four models: DynamiCrafter-VC (VC for fine-tuned from VideoCrafter), FrameBridge-VC, FrameBridge-VC-FS, FrameBridge-VC-NC. **The first two models use the same base model (i.e., VideoCrafter) and fine-tuning setups**, and comparisons between them can fairly show the effectiveness of FrameBridge. The last two models are variants of FrameBridge-VC, which we proposed in our initial response to offer possible methods to further improve the dynamic degree of FrameBridge.
> > >
> > > We randomly sample 50 prompts from VBench-I2V and generate videos with the mentioned 4 models. For each group of videos (i.e. the 4 videos generated with the same prompt by different models), participants are asked 2 quetions:
> > >
> > >
> > > (1) Rank the videos according to the dynamic degree. Higher rank (i.e. lower ranking number) corresponds to higher dynamic degree.
> > >
> > > (2) Rank the videos according to the overall quality. Higher rank (i.e. lower ranking number) corresponds to higher quality.
> > >
> > > We recruited 18 participants (which is a reliable setting in this domain [1, 2, 3]) and use Average User Ranking (AUR) as a preference metric (lower for better performance).
> > > |Model|AUR of DD|AUR of overall quality|
> > > |-|-|-|
> > > |DynamiCrafter-VC|2.85|3.04|
> > > |FrameBridge-VC|2.74|**2.26**|
> > > |FrameBridge-VC-FS|**2.12**|*2.34*|
> > > |FrameBridge-VC-NC|*2.29*|2.35|
> > >
> > > The results show that:
> > >
> > > (1) By comparing DynamiCrafter-VC and FrameBridge-VC, FrameBridge does not have inferior dynamic degree, and the overall quality is better.
> > >
> > > (2) By comparing FrameBridge-VC and the two variants, the proposed solutions are effective.
> > >
> > > Meanwhile, we would like to respectfully clarify a possible misunderstanding in the comment. We stated "the much higher DD of SparseCtrl may lead to inferior overall quality", but did not intend to challenge that DD is reliable for measuring dynamic degree and VBench-I2V total score is reliable for measuring overall quality. The results of user study is largely consistent with DD score: DynamiCrafter-VC and FrameBridge-VC has similar dynamic degree, and the dynamic degree of two variants are much higher.
> > >
> > > ## Related to Q1
> > > We would like to provide another explanation of the differences from an empirical perspective. Previous works in diffusion show that, although theoretically equivalent, even linear changes in the parameterization of diffusion process can lead to significant performance gaps in sample quality [5]. The sampling process corresponds to ODE/SDE trajectories, and linear transformations can induce complicated changes in these trajectories, resulting in empirical inequivalence (commonly used samplers are finite-order approximation, which cannot handle these changes losslessly). For bridge models, the start point of sampling process is shifted to a deterministic point closer to target data, and bridge process causes empirically non-trivial changes of sampling trajectories. The primary contribution of our work is also in the empirical domain. We provide experimental evidences demonstrating that bridge models are empirically more suitable for I2V generation, which aligns with intuition, and propose specific techniques to enhance the I2V generation performance.
> > >
> > > ## Related to Q2
> > > **We offer the possible solution of applying LLM planner since we understood the "scene cuts" means aburpt change. To address the concern thoroughly, we conducted a preliminary experiment (due to time constrain, we use limited computational resources) to show that bridge model itself has the ability to deal with scene cuts without incorporating other tools**. We manually construct a dataset by concatenating two random videos and change the middle frames with interpolation of two videos to manually simulate 3 types of scene cuts (both abrupt and smooth) and find that FrameBridge can model these scene cuts after training with the constructed dataset. We show some samples in: https://framebridge-icml.github.io/rebuttal-supplementary/ (**Table 1,2,3**).
> > >
> > > The intension of this preliminary experiment is to show that scene cut does not pose a limitation to bridge model's expressive capacity. Our discussion from the perspective of generative models (i.e. The second part of our reply to Q2 of Reviewer fXNN) is still applicable to this scenario. We also show an example of sampling process, which we hope could offer some intuitive explanations (**Table 4** in the above link). We would like to focus our discussion on this, and more detailed investigation into scene cut generation is another valuable topic beyond the scope of our paper.
> > >
> > > [1] DynamiCrafter
> > >
> > > [2] InstructVideo
> > >
> > > [3] Conditional Image Leakage
> > >
> > > [4] VBench++
> > >
> > > [5] DPM-Solver++

---

### Official Review · Reviewer_fXNN · 2025-03-14

**Overall Recommendation:** 4

**Summary:**

This work focuses on the mismatching issue of diffusion models and I2V generation tasks, and propose FrameBridge, which build a data-to-data generation process with bridge model, making the generation procedure more in line with the frame-to-frames nature of I2V task. For fine-tuning scenario, a SNR-Aligned Fine-tuning is proposed  to take advantage of pretrained diffusion models. For training from scratch, a trained prior model provides initial prior of non-first frames.

## update after rebuttal

I have carefully read the response from authors and the comments of other reviewers. My major concerns about the adaption to multi-shot generation and limited dynamics were also raised by other reviewers. My concerns have been addressed. I will keep my initial rating.

**Claims And Evidence:**

Yes. This work proposes to model the I2V generation task from frame-to-frames to a data-to-data framework based on the schr\"odinger bridge theory, which is implemented by a SNR aware finetuning or an advanced prior. The effectiveness of the proposed bridge based I2V method is validated on various benchmarks across different metrics.

**Essential References Not Discussed:**

No.

**Experimental Designs Or Analyses:**

The proposed method is validated on MSR-VTT, webvid and UCF-101 compared with previous I2V works. The details of these datasets and the evaluation metrics are elaborated in the appendix.

**Methods And Evaluation Criteria:**

Yes. The benchmarks used to validate are commonly used in this field, including the MSR-VTT and UCF-101, which are standard benchmarks for video generation task.

**Other Comments Or Suggestions:**

- In line 168 right side, the $i$ in $(z_T|i, c)$ is not explained.

**Other Strengths And Weaknesses:**

Strengths:
1. This paper identifies a clear mismatch issue between diffusion models and the frame-to-frames I2V task, which is well-motivated. A Schrödinger bridge-based method is proposed to deal with I2V task.
2. The proposed SAF and neural prior are practical and effective to take advantage of pretrained diffusion models and enable efficient from scratch training.
3. The paper is well-written and easy to follow. The details of experiments are elaborated.

Weaknesses:

please refer to the question part.

**Questions For Authors:**

1. I certainly understand the underlying bridge theory and the proposed method for I2V, considering that some modern video generation methods can process multi-shot or multi-scene video generation, in which scenario the content of multiple shots may be much different with the given first frame, dose the hypothesis of starting from repeated first frame via bridge still make sense? Is there any more improvement or consideration of the issue?
1. In the I2V task, repeating the first frames across the temporal dimension may influence the final motion of the generation result, limiting achieving high dynamic motions. Could the authors discuss about this issue?

**Relation To Broader Scientific Literature:**

This work is mainly related to I2V video generation task, but implemented based on the schr¨odinger bridge theory. This theory has been explored these years for synthetic tasks like style transfer, I2I generation, etc. The bridge theory is more in line with these data-to-data task, compared with the commonly used diffusion method.

**Theoretical Claims:**

Yes, the theoretical claims in the appendix is reviewed, including the analysis of parameterization of the bridge optimization objectives and the analysis of SAF and the proposed prior for training from scratch.

---

> ### Author Rebuttal · Authors · 2025-04-01
>
> Dear Reviewer fXNN,
>
> We sincerely appreciate your acknowledgement of our methods and proposed techniques as "well-motivated", "practical and effective". We are happy to engage in a thorough discussion and hope it will address your concerns and questions. Tables and demos are provided in https://framebridge-icml.github.io/rebuttal-demo-page/.
>
> ## Reply to Q2
>
> **Firstly, we will offer some empirical solutions to improve the dynamic degree of FrameBridge.** The limited dynamic degree is a well-studied issue for diffusion I2V models [2,3,4]. We adapt two techniques used in diffusion I2V models to FrameBridge: (1) Add frame-stride condition; (2) Add noise to image condition. Demonstrated by our experments on FrameBridge-VideoCrafter, we find that these techniques are also effective for FrameBridge, and can **improve the DD score from 35.77 to 48.62**, which is much higher than the diffusion counterpart (38.69 for DynamiCrafter). (See **Table 2,7,8**. -MI for Motion-Improved).
>
> **Secondly, we would like to mention that the potential limited dynamic degree is an empirical issue instead of the inherent weakness of bridge-based methods.** Just as diffusion models, if we have the perfect bridge score function and a perfect sampler for the backward bridge SDE (Eq. 5 in the paper), we can sample from the true data distribution, which is a mathematically well-established conclusion in [6]. It may be counterintuitive. To understand it **more intuitively (but not that rigorously)**, we would like to provide discussions from three aspects:
>
> 1. **Marginal:** Consider the marginal $z_t=a_t z_0 + b_t z_T + c_t \epsilon$, where $a_0=b_1=1, a_1=b_0=c_0=c_1=0$. Different from the diffusion process, the coefficient of noise $c_t$ first increases then decreases when $t$ varies from $T$ to $0$ in the sampling process. So, the latent $z_t$ **will continuously become noisier in the first half part of sampling process**.
>
> 2. **Forward-Backward SDE:** In [6] a forward diffusion process
> $$
> \mathrm{d}z_t = f(t)z_t \mathrm{d}t + g(t)\mathrm{d}w,
> $$
> is used to construct the forward bridge process
> $$
> \mathrm{d}z_t=[f(t)z_t + g(t)^2 \nabla_{z_t} \log p_{T,diff}(z_T|z_t)]\mathrm{d}t+g(t)\mathrm{d}w,
> $$
> Compared with diffusion process, the additional term $g(t)^2 \nabla_{z_t} \log p_{T,diff}(z_T|z_t)$ gradually "pushes" $z_t$ to the value $z_T$ and finally the prior $z_T$ will become a fixed point. Reversely, compared with diffusion backward process, the backward bridge process will additionally "pull" $z_t$ away from $z_T$.
>
> 3. **Comparison between bridge model and inference-time noise manipulation:** In FrameInit, the prior $z_T$ is constructed by combining low-frequency information from a static video with Gaussian noise. (Similar to FrameBridge at first glance.) Then, $z_T$ is denoised with a **trained diffusion model**. Although $z_T$ has been influenced by the low-frequency components of a static video, the diffusion process will sample from it in the same way as a Gaussian noise, **since the training process of diffusion model is "ignorant" of the noise manipulation and still learns the score function of a standard diffusion process $z_t=\alpha_tz_0+\sigma_t\epsilon$, which may cause train-test mismatch and limit the dynamic degree**. (This may also explain why the Dynamic Degree score of ConsistI2V is lower.) However, for bridge model, we change the prior $z_T$ **along with the diffusion process. The model is "aware" of that and will learn the score function of a bridge process $z_t=a_0z_0+b_tz_T+c_t\epsilon$ with bridge score matching loss.**
>
> ## Reply to Q1
>
> This is a valuable question, and multi-shot/scene I2V generation is a highly demanding task for I2V generation without mature benchmarks or baselines currently. It will also be challenging for FrameBridge as repeating the first frame as prior may not be a proper design. Inspired by RPGDiffusion [8] and other previous works about long video generation [9,10, 11] , we propose a possible hierarchical solution as below:
>
> 1. Use LLM planner to divide the whole video into several sub-video clips $v_1, v_2, ..., v_M$, and generate image prompts for the start of each clip.
>
> 2. Generate the start of each clip $i_1, i_2,..., i_M$ with a T2I diffusion model.
>
> 3. For $v_t, 1 \leq t \leq M - 1$, construct a neural prior by interpolating between $i_t, i_{t + 1}$ and train a FrameBridge for video interpolation (which is also a typical task for I2V generation [12]) to generate it. Generate $v_M$ with the static prior constructed by $i_M$ using FrameBridge.
>
> Our consideration is to generalize the neural prior with LLM planner and use FrameBridge to leverage the locally reliable prior.
>
> ## Other Discussions
>
> The line 168 is a typo and should be $(z_T | z^i, c)$. We thank the reviewer to point it out and will revise it later.
>
> Due to word limit, we can not elaborate on all the details and please feel free to tell us if you still have any concern.
>
> References: See response to Reviewer g6Xo.

---

### Decision · Program_Chairs · 2025-05-01

**Decision:**

Accept (poster)

**Comment:**

This paper presents FrameBridge, a Schrödinger Bridge-based approach for image-to-video generation that reframes the task as data-to-data synthesis. While initial concerns regarding the neural prior's blurriness and inconsistent baselines were raised, the authors addressed these via clarifying design choices (e.g., concatenation preserves details) and supplemental experiments with unified settings. The method’s effectiveness is validated across benchmarks (MSR-VTT/UCF-101) with metrics (FVD, CLIPSIM), and new 32-frame results in the rebuttal alleviate concerns about long-video limitations . Though technical contributions appear incremental (bridges + diffusion), the SNR-aware finetuning (SAF) and theoretical grounding provide actionable insights. Thus, I recommendation to accept this paper.